# Unravelling the spatial directionality of urban mobility

Pengjun Zhao ®[1,2] ✉, Hao Wang ®[2], Qiyang Liu ®[2] ✉, Xiao-Yong Yan ®[3] ✉ & Jingzhong Li[4]

As it is central to sustainable urban development, urban mobility has primarily been scrutinised for its scaling and hierarchical properties. However, traditional analyses frequently overlook spatial directionality, a critical factor in city centre congestion and suburban development. Here, we apply vector computation to unravel the spatial directionality of urban mobility, introducing a two-dimensional anisotropy-centripetality metric. Utilising travel data from 90 million mobile users across 60 Chinese cities, we effectively quantify mobility patterns through this metric, distinguishing between strong monocentric, weak monocentric, and polycentric patterns. Our findings highlight a notable difference: residents in monocentric cities face increasing commuting distances as cities expand, in contrast to the consistent commuting patterns observed in polycentric cities. Notably, mobility anisotropy intensifies in the outskirts of monocentric cities, whereas it remains uniform in polycentric settings. Additionally, centripetality wanes as one moves from the urban core, with a steeper decline observed in polycentric cities. Finally, we reveal that employment attraction strength and commuting distance scaling are key to explaining these divergent urban mobility patterns. These insights are important for shaping effective policies aimed at alleviating congestion and guiding suburban housing development.

Human mobility, a critical factor in fields ranging from disease control[1–3] and traffic forecasting[4,5] to urban planning[6], has seen significant advancements due to the integration of multi-source data such as currency flow[7], global positioning system (GPS)[8], and mobile phone records[9,10]. These data sources offer unprecedented spatiotemporal resolution, enhancing our understanding of individual and population mobility[11,12]. Early research primarily excavated basic mobility characteristics and developed models to reproduce them. At the individual level, studies have uncovered universal scaling laws, such as the fat-tailed distribution of jump sizes, explained by continuous-time random walk[7]. At the population level, the emphasis mainly focuses on modelling inter-location mobility flows, leading to universal models such as the radiation model[13] and the population-weighted opportunities model[5,12].

Recently, the exploration of human mobility has pivoted towards utilising quantitative metrics to delineate the overarching characteristics of urban mobility. These metrics predominantly rely on non-spatial data, drawing from the numerical insights of origin-destination (OD) flow matrices. For instance, Louail et al.[14] utilised the non-spatial OD matrix to create a hotspot flow matrix, assessing the flow ratios between work and residential hotspots across different tiers. Their analysis demonstrated that larger cities often show reduced flow ratios from key residential to workplace hotspots. Similarly, Bassolas et al.[15] introduced a flow hierarchy, again derived from non-spatial OD matrix data, to evaluate the flow ratios among locations with comparable hotspot rankings, thereby unveiling the layered nature of urban mobility. Their research revealed a pronounced stratification in the

[1]College of Urban and Environmental Sciences, Peking University, Beijing 100871, China. [2]School of Urban Planning and Design, Shenzhen Graduate School, Peking University, Shenzhen 518055, China. [3]School of Systems Science, Beijing Jiaotong University, Beijing 100044, China. [4]College of Urban and Environmental Sciences, Xuchang University, Xuchang 461000, China. ✉e-mail: pengjun.zhao@pku.edu.cn; tsql@pku.edu.cn; yanxy@bjtu.edu.cn

mobility structures of cities across Asia and Africa, in contrast to the more homogeneous patterns observed in Europe and the Americas, and notably less so in Oceania. These indicators, focusing on the non-spatial directionality of urban mobility, diverge from traditional studies on urban spatial structure based on population density, infrastructure, and employment distribution[16–20], offering new insights into urban spatial configurations. However, despite advancements in understanding urban mobility through non-spatial directionality, spatial directionality—the equilibrium characteristics and orientation tendencies of flows across various spatial directions towards or away from specific urban landmarks, typically the city centre—remains underexplored (Fig. 1 and Supplementary Fig. 1).

The spatial directionality of urban mobility is indispensable for revealing complex urban mobility patterns and dynamics. Sole reliance on non-spatial OD data inadequately represents the nuanced mobility of urban residents. For instance, should the city's central business district (CBD) relocate to the periphery, traditional OD matrix analyses would fail to detect any shifts in mobility patterns. Yet, factoring in spatial directionality uncovers profound changes: resident mobility patterns, initially centred towards the CBD, would markedly pivot towards the new location, fundamentally transforming the city's

mobility landscape. This example underscores the limitations of non-spatial OD data, which, devoid of spatial context, cannot capture the essence of mobility changes. Conversely, spatial directionality offers a lens through which these changes become apparent, proving essential for a nuanced understanding of urban dynamics and how residents interact with their urban environment. Despite its significance, there remains a gap in metrics that fully integrate spatial directionality into the quantification of urban mobility.

We aim to bridge this crucial gap by quantifying the spatial directionality of urban mobility, employing vector computation that integrates both the magnitude and the direction of mobility flows. This analysis utilises data from approximately 90 million mobile phone users across 60 Chinese cities in August and November 2019 (Methods). These cities predominantly consist of directly administered municipalities and provincial capitals with superior economic development, embodying the mobility characteristics of China's most representative major cities. We introduce the concept of the population mobility vector (PMV)[21], which naturally leads to the development of two key metrics: anisotropy and centripetality. These metrics adeptly capture the imbalances in direction distribution and the orientation towards the city centre in mobility flows. Employing these

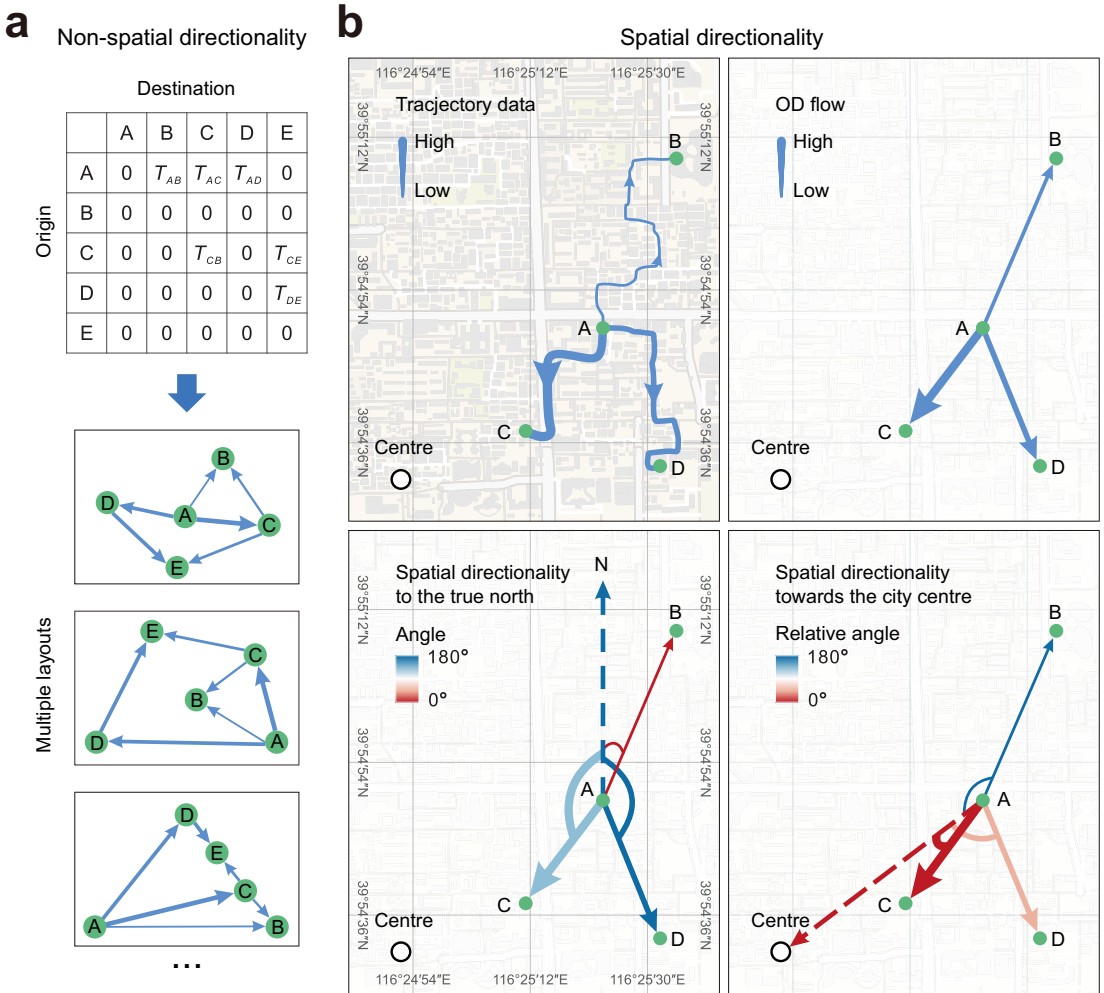

**Fig. 1 | Comparison of non-spatial and spatial directionality. a** Shows an origin-destination (OD) matrix for locations A to E, illustrating non-spatial directionality where the lack of spatial information allows for various potential flow directions. **b** Shows spatial directionality, starting with a diagram of the locations and potential paths (top left), followed by actual flow directions with arrow thickness indicating volume (top right). The bottom-left diagram shows spatial directionality relative to true north, with colours shifting from red (aligned with true north) to blue (opposite direction). Similar colours for flows from A to C and A to D indicate parallel spatial orientations. The bottom-right diagram visualises spatial directionality towards the city centre, with red indicating closer alignment and blue the opposite. This reveals that the flow from A to C aligns more towards the city centre than the flow from A to B.

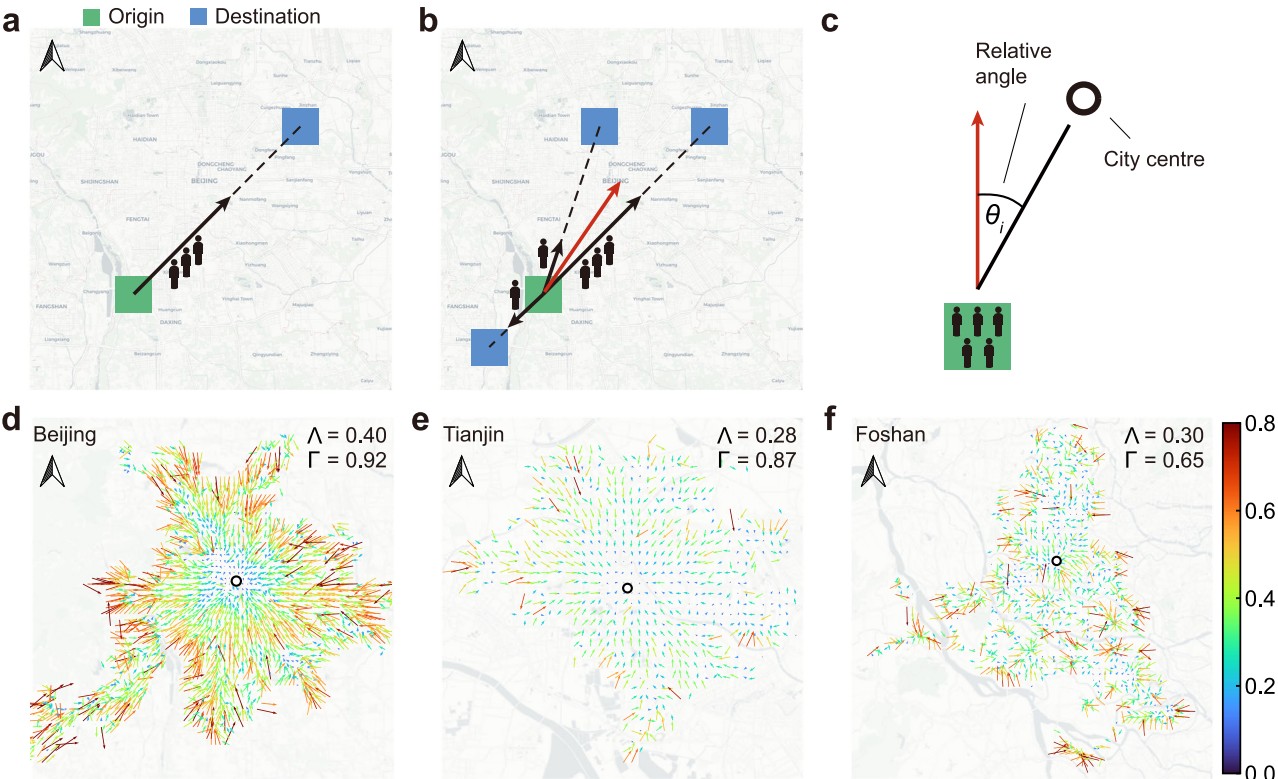

**Fig. 2 | Population mobility vector (PMV). a** Schematic diagram showing the method to vectorise the mobility flow on a Cartesian plane with the y-axis along the geographical meridian and pointing towards true north. The direction of the unit outflow vector (black arrow) is defined from the origin (*i*, green) to the destination (*j*, blue), and the magnitude is defined as the relative flow (i.e., the flow from *i* to *j* normalised by the outflow $O_i$). **b** Definition of PMV. The origin *i* has three destinations, and each flow from *i* to *j* can be represented as a vector using the method shown in (**a**). Taking a vector sum of these vectors, we obtain the PMV $\vec{T}_i$ (red arrow) on this Cartesian plane. **c** Definition of the relative direction. The relative direction ($\theta_i$) of the PMV is defined as the direction referenced to the city centre (white dot, Methods), which can be calculated as the angle between the direction of the PMV (red arrow) and the direction to the city centre (black line). **d**–**f** Maps of PMVs for **d** Beijing, **e** Tianjin, and **f** Foshan. The length and colour of the arrow are proportional to the magnitude of the PMV. The white dot represents the city centre. Vector basemap and map tiles were provided by CartoDB under a Creative Commons licence CC BY 4.0. Source data are provided as a Source Data file.

metrics, we can precisely quantify the spatial directional characteristics of cities, enabling us to classify urban areas and dissect their distinct mobility patterns. Our study extends to examining the correlations between average commuting distances and city sizes across varied mobility patterns. We then delve into a comprehensive spatial analysis of these two metrics, exploring the directional features of different mobility patterns within the urban fabric. Furthermore, we scrutinise the temporal dynamics of mobility directionality, investigating how these characteristics fluctuate throughout the day for different mobility patterns. In the final phase of our research, we construct a microscopic model to elucidate the diverse characteristics observed in urban mobility. Our work not only offers an approach for quantifying and categorising the overarching traits of urban mobility, but also sheds light on their spatiotemporal variations. This contributes to urban and transport planning, providing an analytical framework for understanding and managing urban mobility.

## Results

### Anisotropy and centripetality of mobility flows

To quantify the spatial directionality of urban mobility, we introduce two metrics. Initially, we vectorise the flow from origin to destination $T_{ij}\vec{u}_{ij}$, where $T_{ij}$ represents the flow from origin *i* to destination *j*, and $\vec{u}_{ij}$ is the unit vector from origin *i* to destination *j* on a Cartesian plane with the y-axis along a geographical meridian and pointing towards true north. To mitigate the effects of varying outflow volumes across different locations[15,21,22], we normalise the obtained vectors using the total outflow $O_i$ from each location, resulting in a unit outflow vector (Fig. 2a). Subsequently, we aggregate all unit outflow vectors for

location *i* through vector summation (Fig. 2b) and define this sum as the PMV on this Cartesian plane

$$\vec{T}_i = \sum_{j \neq i} \frac{T_{ij}}{O_i} \vec{u}_{ij}. \tag{1}$$

PMV emerges as a comprehensive result of aggregating the characteristics of all flows originating from a specific location *i*. It encapsulates the collective mobility patterns of all flows emanating from location *i*, reflecting an integrated mobility characteristic of the said point of origin.

Further, based on the magnitude and direction of the PMV relative to the city centre (Fig. 2c), we define two key metrics of urban mobility: anisotropy ($\Lambda$) and centripetality ($\Gamma$) (Methods). These metrics respectively quantify the uneven distribution of urban mobility across different directions and the degree of orientation towards the city centre, embodying the spatial directional characteristics of mobility within urban areas (Methods). Higher anisotropy indicates an unequal distribution of flows across various directions, with traffic concentrating in certain specific directions, whereas higher centripetality suggests that flows predominantly move towards the city centre. Thus, significant anisotropy and centripetality values could imply that residents' mobility is directed towards the city centre, indicating a monocentric mobility pattern. In contrast, lower values of both metrics would suggest random mobility directions across the city, indicative of a polycentric mobility pattern. Here, we calculate the anisotropy and centripetality of multiple cities during the typical morning rush hour at 7 a.m., which accurately reflects the dynamics

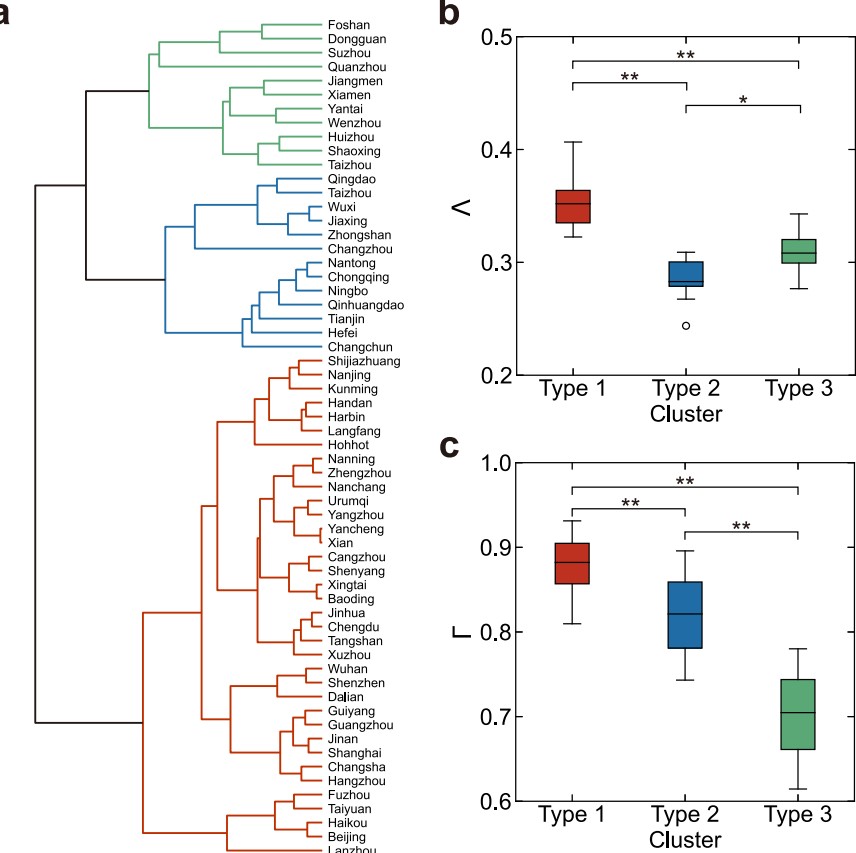

**Fig. 3 | Urban mobility patterns. a** Hierarchical clustering of cities based on their anisotropy and centripetality of commuting flows. The dendrogram presents the hierarchical clustering of the $n = 60$ cities studied in the research, and the colours illustrate a three-cluster partitioning. The various colours correspond to different types of cities: Type 1 (strong monocentric, red, $n = 36$ cities), Type 2 (weak monocentric, blue, $n = 13$ cities) and Type 3 (polycentric, green, $n = 11$ cities). **b** Comparison of anisotropy across Type 1−3 cities. Two-sided $t$-tests were conducted with Bonferroni correction for multiple comparisons. Type 1/2:

$P = 9.43 \times 10^{-13}$, Type 2/3: $P = 0.003$, Type 1/3: $P = 7.96 \times 10^{-7}$. **c** Same as in (**b**) but for centripetality. Type 1/2: $P = 2.40 \times 10^{-5}$, Type 2/3: $P = 4.70 \times 10^{-6}$, Type 1/3: $P = 2.60 \times 10^{-17}$. For the box plots in (**b**, **c**), the central mark indicates the median, and the bottom and top edges of the box indicate the 25th and 75th percentiles, respectively; the whiskers extend to the most extreme data points within 1.5 times the interquartile range from the bottom or top of the box, and all more extreme points are plotted individually using a circular symbol. $^*P < 0.05$; $^{**}P < 0.001$. Source data are provided as a Source Data file.

between residences and workplaces (Supplementary Table 4). To elucidate the characteristics of anisotropy and centripetality, we showcase examples from Beijing, Tianjin, and Foshan (Fig. 2d−f). Comparatively, Beijing exhibits higher anisotropy and centripetality ($\Lambda = 0.40$, $\Gamma = 0.92$), showcasing a monocentric mobility pattern. Foshan displays lower anisotropy and centripetality ($\Lambda = 0.30$, $\Gamma = 0.65$), indicating a polycentric mobility pattern. Meanwhile, Tianjin shows lower anisotropy but higher centripetality ($\Lambda = 0.28$, $\Gamma = 0.87$), possibly representing a transitional pattern between the two. Our approach offers a coarse-grained representation of mobility flows, simplifying detailed descriptions while retaining meaningful information on the spatial directionality of urban mobility.

**Clustering of mobility patterns**

The anisotropy and centripetality metrics effectively measure commuting flows in various urban configurations during the typical morning rush hour. Cities are plotted in an anisotropy-centripetality space, and distances between points assess mobility directionality similarities[14]. Hierarchical clustering divides cities into three distinct clusters (Fig. 3a) with notable differences in anisotropy and centripetality (Fig. 3b, c). The robustness of this classification is confirmed against various clustering approaches, unaffected by proximity measures in hierarchical algorithms (Supplementary Note 5 and Supplementary Fig. 6).

Furthermore, the classification significantly impacts urban mobility patterns. Type 1 cities (Fig. 3a, red) exhibit high anisotropy

and centripetality (Fig. 3b, c), indicative of a strong monocentric pattern. This pattern is characterised by a dominant CBD and sub-urban residential areas, resulting in high commuting flows along radial directions towards the centre[23,24]. In contrast, Type 3 cities (Fig. 3a, green) display low levels of both anisotropy and centripetality (Fig. 3b, c), reflecting a polycentric pattern. This pattern is characterised by intermixed residential and workplace areas, leading to more isotropic and centrifugal commuting behaviours. Type 2 cities (Fig. 3a, blue) show a combination of low anisotropy and high centripetality (Fig. 3b, c), indicative of a weak monocentric pattern. In these cities, suburban development leads to a relatively balanced distribution of residences and workplaces, resulting in more isotropic commuting behaviour. However, the CBD still possesses many job opportunities and thus attracts significant inward commuter flows[24]. This classification is visually corroborated by distinct travel characteristics within each city type (Supplementary Note 6 and Supplementary Fig. 7).

Note that the concept of centripetality presumes the existence of a distinctive city centre. Cities with strong and weak mono-centric patterns have a clear centre towards which most commuting flows are directed. Polycentric cities, however, feature multiple centres. Following our urban centre identification methodology (Methods), the primary centre in polycentric cities is considered the most influential among these, significantly impacting city-wide flows.

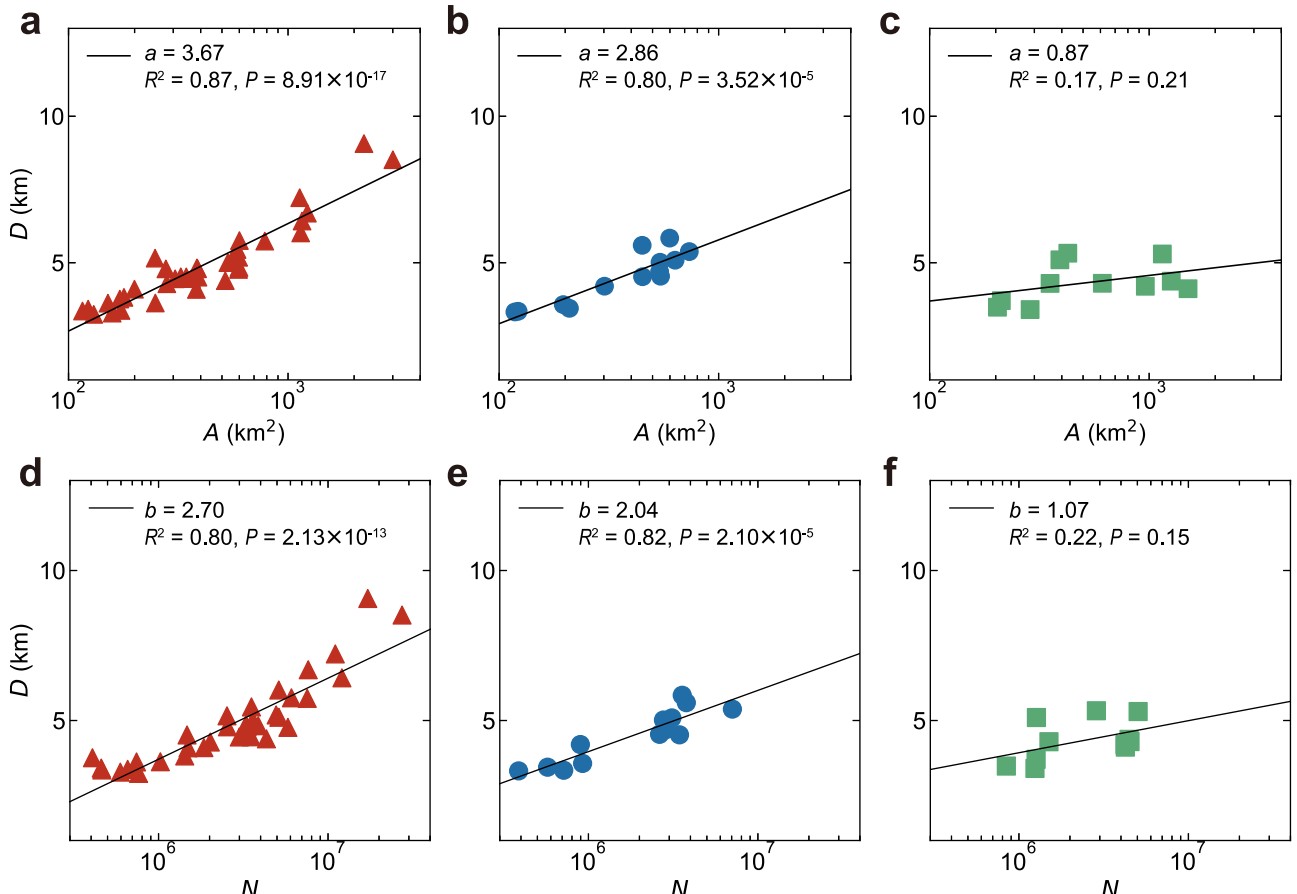

**Fig. 4 | The relationship between the average commuting distance and city size. a–c** Average commuting distance versus urban area size for **a** strong monocentric cities, **b** weak monocentric cities and **c** polycentric cities. The solid line indicates a logarithmic function fit ($D \sim \log A^a$) for which the parameter is provided in each panel. **d–f** Average commuting distance versus urban population size for **d** strong monocentric cities, **e** weak monocentric cities and **f** polycentric cities. The solid line indicates a logarithmic function fit ($D \sim \log N^b$) for which the parameter is provided in each panel. A linear regression $t$-test was conducted to determine whether the slope of the regression line differed significantly from zero. Source data are provided as a Source Data file.

## Commuting distance of different patterns

We explore the link between average commuting distance (Methods) and city size to understand urban mobility patterns. The average commuting distance is a key indicator of mobility efficiency, fuel consumption, and $CO_2$ emissions[25]. To represent city size, we employ both urban area size and urban population size. Our analysis reveals a correlation between the average commuting distance ($D$) during the typical morning rush hour and urban area size ($A$) across three distinct mobility patterns, as depicted in Fig. 4a–c. In monocentric cities, there is a strong positive correlation between commuting distance and urban area size (Fig. 4a, b; $R^2 = 0.87$, $P < 0.001$ and $R^2 = 0.80$, $P < 0.001$), aligning with the expectation that larger cities generally necessitate longer commuting distances[14]. Most of these cities have expanded their urban areas by gradually spreading out from the CBD. This growth pattern often leads to the expansion of residential areas, while most workplaces remain in their original locations. Therefore, the average commuting distance increases rapidly with an urban area under the combined effects of urban expansion and the separation of residences and workplaces, with strong monocentric cities experiencing faster increases than weak monocentric cities.

In contrast, polycentric cities (Fig. 4c; $R^2 = 0.17$, $P = 0.21$) show no significant correlation between commuting distance and urban area size, indicating different growth patterns from monocentric cities. Typically formed by merging smaller settlements, polycentric cities have mixed residences and workplaces, resulting in relatively stable commuting distances despite urban expansion. Additional analysis on urban population size $N$ (Fig. 4d–f) shows that while commuting distances in monocentric cities increase with a growing urban population, such trends are relatively stable in polycentric cities.

These observations underscore the efficacy of our approach in quantifying spatial directionality to differentiate between mobility patterns. Crucially, they unveil how the average commuting distance in cities with differing mobility patterns varies with city size, indicating the pace of change. In monocentric cities, larger cities have longer average commuting distances, whereas in polycentric cities, average commuting distances tend to stabilise as city size increases.

## Spatial variations in anisotropy and centripetality

Here we delve into the mesoscopic scale spatial distribution of anisotropy and centripetality during the typical morning rush hour across cities, building on a comprehensive city-scale urban mobility analysis. Inspired by Bertaud and Malpezzi's framework[18], we delineate cities into concentric rings around the urban centre (Fig. 5a), reflecting the multi-ring road networks developed in response to urban expansion, demographic growth, and transport needs. These networks, particularly the quintessential five-ring structure, exemplify the urban form of many Chinese cities. This structure serves as a model for segmenting all cities, even those without a literal five-ring configuration. By adopting a uniform ring zone classification based on equal trip generation, we facilitate comparative analysis across diverse urban sizes. Here we use $l = 5$ for all cities, while our results are robust against changes in this value (Supplementary Note 7 and Supplementary Fig. 8). Each level's anisotropy and centripetality were then calculated (Supplementary Note 8), revealing two key patterns.

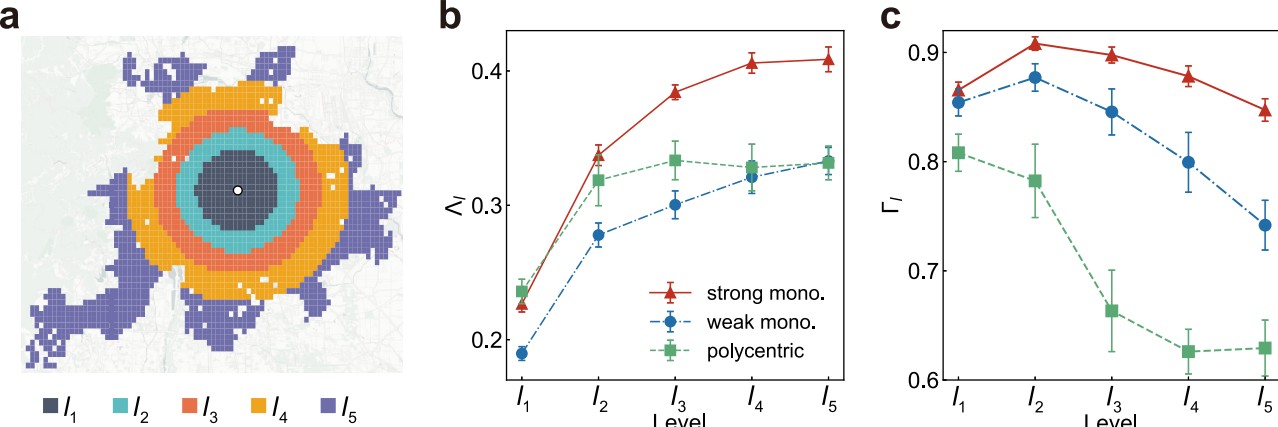

**Fig. 5 | The spatial distribution of anisotropy and centripetality. a** Illustration of the spatial hierarchical structure in Beijing. Each colour corresponds to a space level, and trip generation of different levels is equal. Illustration of the spatial hierarchical structure in Beijing, where each colour denotes a distinct spatial level, with equal trip generation across levels. This equality is defined by the premise that the total outflow originating from the cells within each colour-coded area is identical. Vector basemap and map tiles were provided by CartoDB under a Creative Commons licence CC BY 4.0. **b, c** The **b** anisotropy and **c** centripetality of each spatial level for strong monocentric ($n = 36$), weak monocentric ($n = 13$) and polycentric cities ($n = 11$). Error bars correspond to standard errors and centre values correspond to the means. Symbols and lines refer to various city types. Source data are provided as a Source Data file.

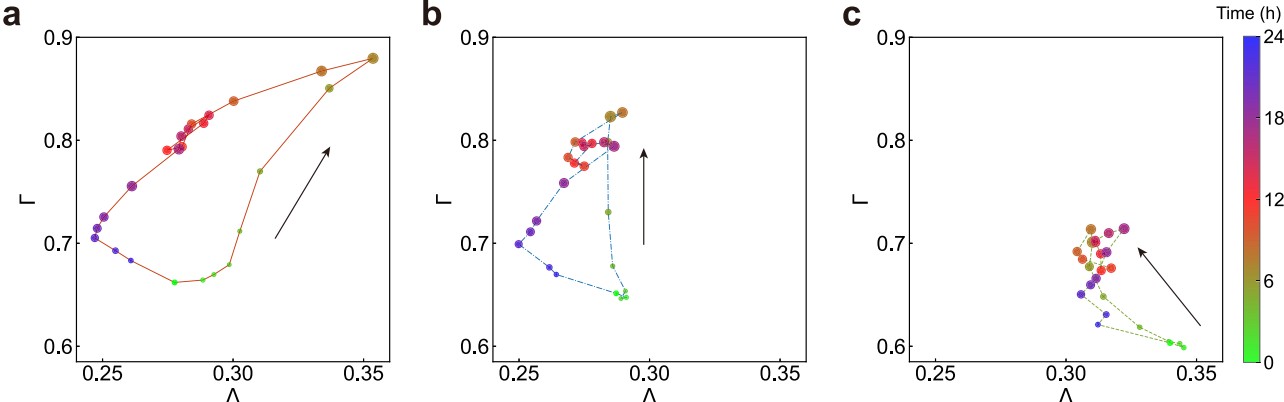

**Fig. 6 | The temporal dynamics of anisotropy and centripetality. a–c** Hourly anisotropy and centripetality of urban mobility for **a** strong monocentric cities ($n = 36$), **b** weak monocentric cities ($n = 13$) and **c** polycentric cities ($n = 11$). Symbol sizes indicate the relative trip generation volume per hour. Symbol colours denote different hours. Green and blue signify early morning and late night, with yellow and red indicating midday. Source data are provided as a Source Data file.

Firstly, anisotropy consistently escalates with spatial level across all city types (Fig. 5b and Supplementary Fig. 9a–c). This trend suggests a weaker imbalance in commuting flow directions within urban cores, with a marked increase in the peripheries. One main reason for this may be that job opportunities are not distributed evenly across space, with the urban core offering more balanced options in each direction. Notably, polycentric cities show relatively stable anisotropy beyond Level $l_2$, indicating that the direction distribution of job opportunities is approximately equal in their periurban areas. Contrastingly, strong monocentric cities exhibit the most pronounced increase in anisotropy (80.3%), followed by weak monocentric (75.5%) and polycentric cities (40.5%). The varying degrees of increase in anisotropy indicate that the direction distribution of job opportunities in the periurban area of strong monocentric cities is the most imbalanced compared to their urban cores, followed by weak monocentric cities, and finally polycentric cities.

Secondly, centripetality displays a systematic decline with increasing spatial levels across all city types (Fig. 5c and Supplementary Fig. 9d–f). This finding indicates a stronger orientation towards the city centre in urban cores, diminishing in peripheral areas. It points to a core-periphery structure within cities[22], where the urban core's gravitational pull on commuting flows wanes as one moves outward. Intriguingly, monocentric cities manifest relatively lower centripetality at Level $l_1$, implying less city centre-oriented mobility within these zones. This is because these two types of monocentric cities have predominant CBDs within Level $l_1$, which provide many job opportunities. Therefore, residents within Level $l_1$ have more freedom of choice in their workplaces, resulting in a more diverse mobility orientation. The decline in centripetality is lowest in strong monocentric cities (2.1%), followed by weak monocentric (13.2%) and polycentric cities (22.1%). This gradient suggests that the urban core's attractiveness is highest in strong monocentric cities, diminishing in weak monocentric cities, and least in polycentric cities.

The investigation of this spatial hierarchical structure further corroborates the efficacy of our method in quantifying spatial directionality to distinguish diverse mobility patterns. Moreover, this mesoscale study captures the variations in the spatial directionality of mobility within urban spaces, thereby offering more scientific, detailed and actionable strategic references for urban planning and management.

## Temporal dynamics of anisotropy and centripetality

Next, we analyse the diurnal variations in the anisotropy and centripetality of urban mobility, focusing on a 1-day cycle at the city scale. Figure 6 shows the average hourly variations for the three city types (see Supplementary Fig. 10 for the details about individual cities).

Notably, strong monocentric cities exhibit the most pronounced fluctuations in both anisotropy and centripetality, surpassing those in weak monocentric and polycentric cities (Supplementary Table 5). In particular, both types of monocentric cities show consistent patterns during off-peak hours (9 to 23 h), but they diverge significantly during morning peak hours (4 to 8 h), in contrast to the distinct daily variation observed in polycentric cities.

Furthermore, despite the different values of anisotropy and centripetality, the trend is the same for the three types of cites during off-peak hours, suggesting that the temporal variation trend is an intrinsic and universal property of urban mobility during this period. For example, the anisotropy and centripetality exhibit minimal fluctuation from 9 to 17 h, indicating stability in the spatial directionality of urban mobility. From 18 to 21 h, corresponding with typical homeward commutes, both metrics decrease, reflecting a shift towards more isotropic and centrifugal movement patterns. Post 22 h, as nighttime activities commence, anisotropy increases while centripetality lessens, denoting a change towards more anisotropic and centrifugal leisure flows.

During morning peak hours, the city types differ significantly in their temporal anisotropy and centripetality patterns (Supplementary Table 6). Strong monocentric cities show the most substantial increase in centripetality (23.6%), followed by weak monocentric (22%) and polycentric cities (15.4%). This gradient suggests that the highest urban core attractiveness is in strong monocentric cities. In terms of anisotropy, strong monocentric cities experience an increase throughout the peak period, whereas it remains relatively stable in weak monocentric cities and decreases in polycentric cities. This trend indicates that the dynamic state of the direction distribution of job opportunities in strong monocentric cities becomes increasingly imbalanced with the increase in trip volume during this period, while weak monocentric cities maintain stability and polycentric cities demonstrate a more balanced distribution.

Remarkably, polycentric cities exhibit higher anisotropy than weak monocentric cities throughout the day, which is attributed to consistent directional mobility within their multiple subcentres. For instance, Junan in Foshan displays significant anisotropy during the morning peak, influenced by the gravitational pull of its business centres on the surrounding rural areas (Supplementary Fig. 11). Taken together, these findings validate the classification of city types based on anisotropy and centripetality, offering insights into their distinct spatiotemporal mobility dynamics.

## Microscopic model of mobility patterns

Our empirical analysis discerns three distinct mobility patterns based on the anisotropy and centripetality of commuting flows. To elucidate these observations mechanistically, we construct a model inspired by conventional methodologies in spatial economics and social physics[26,27], which typically leverages workplace and residence choice behaviours to articulate urban spatial structures. In this context, we introduce the random workplace and residence choice (RWRC) model. The model operates on an $l \times l$ lattice in 2D Euclidean space, representing potential locations. It begins with an individual selecting the lattice centre as both workplace and residence[28,29], and evolves by adding new individuals at each time step (Fig. 7a), each making workplace and residence choices[13,27]. Workplace choice by an individual for location $j$ occurs with probability

$$P_j \propto N_j^{\alpha}, \tag{2}$$

where $N_j$ is the sum of the residential and working populations and represents the active population at $j$, serving as a proxy for employment opportunities[28]. This workplace selection rule is an assumption known as preferential attachment in network and social science[10,30]. The parameter $\alpha \geq 0$ quantifies the attractiveness of population density

for employment. Residence selection for location $i$ ($i \neq j$) is governed by a probability inversely proportional to the distance $d_{ij}$, modulated by a scaling parameter $\beta > 0$, reflecting the increase in commuting costs with distance[31,32]

$$Q_{ij} \propto e^{-d_{ij}/\beta}. \tag{3}$$

Numerical simulations were conducted with a population of $10^4$ on a $50 \times 50$ grid (Methods). The employment attraction strength $\alpha$ was varied from 0 to 2 in increments of 0.1, while the commuting distance scale $\beta$ was increased from 0.01 in steps of 0.1 until anisotropy and centripetality became stable. The results (Fig. 7b) reveal that when both $\alpha$ and $\beta$ are relatively large (triangle), the majority of added individuals tend to choose the central areas of the simulation space as their workplaces and reside in the more distant peripheral regions. As a result, commuting flows exhibit strong anisotropy and centripetality, as exemplified in Fig. 7c, aligning with strong monocentric patterns seen empirically (Fig. 2d). Conversely, when both $\alpha$ and $\beta$ are relatively small (square), the majority of added individuals tend to choose workplaces uniformly and reside in proximity. As a result, commuting flows exhibit weaker anisotropy and centripetality, as exemplified in Fig. 7e, resonating with polycentric patterns seen empirically (Fig. 2f). Moderate values of $\alpha$ and $\beta$ correspond to low anisotropy and high centripetality (diamond), as exemplified in Fig. 7d, akin to weak monocentric patterns (Fig. 2e). These findings indicate that the RWRC model successfully replicates observed commuting characteristics, suggesting that the interplay between employment attraction strength and commuting distance scale fundamentally shapes urban commuting patterns. It bridges the gap between the microscopic workplace-residence choice behaviour of individuals and the macroscopic patterns of urban mobility. This connection deepens our understanding of urban mobility patterns and provides policymakers with a robust theoretical basis. Consequently, it facilitates the formulation of precise and impactful strategies to optimise urban commuting patterns, thereby aiding in the development of more sustainable and livable urban environments.

## Discussion

With the growing accessibility of large-scale mobility data, there has been a surge in human mobility research[11]. Early research primarily focused on basic mobility aspects, such as jump length distributions and inter-location flows, leading to models reproducing these mobility patterns[7–13]. Recent research has evolved to examine the overall characteristics of urban mobility and their impact on city liveability and large-scale events[14,15,33,34]. However, a key aspect of human mobility, spatial directionality, is often overlooked. This paper addresses this critical need for formulating a method that can quantify the spatial directionality associated with urban mobility. We use vectors to describe human mobility and reduce the OD matrix into two metrics, namely, anisotropy and centripetality. These metrics allow for the assessment of directional imbalances and orientations in mobility flows, providing a comprehensive view of urban mobility and insights into urban rhythm[23].

Our metrics distinguish themselves from traditional methods by effectively capturing the spatial directionality of urban mobility. Traditional approaches, such as the hotspot flow matrix[14] and flow hierarchy[15] outline mobility organisation through flow proportions between activity hotspots but fail to capture the spatial complexities inherent in human mobility. Drawing inspiration from the route geometric metric[22], which considers direction, orientation, and length of travel routes, we proposed the PMV to capture these elements, including orientation relative to the city centre. The PMV conceptually aligns with the vector field approach to recurrent mobility[21]. However, our approach diverges significantly in its underlying research

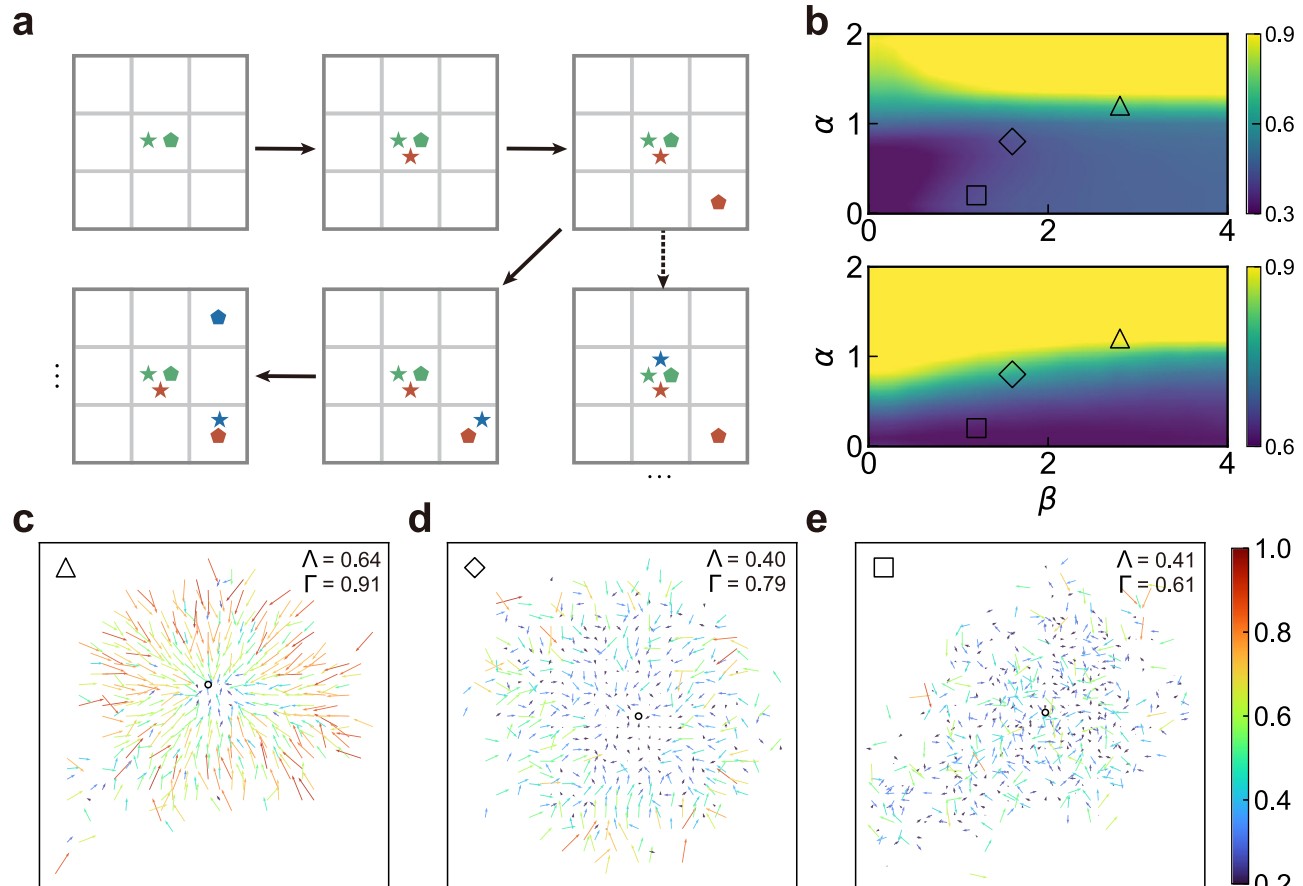

**Fig. 7 | The random workplace and residence choice (RWRC) model. a** Schematic representation of the RWRC model. Individuals are shown as different coloured symbols. The first (green) individual selects the central cell as the workplace (star) and residence (pentagon), as shown in the upper-left panel. Subsequently, the second (red) and third (blue) individuals make their workplace (star) and residence (pentagon) selections in accordance with the criteria outlined in equation (2) and equation (3). The solid arrow represents one choice outcome, whereas the dashed arrow denotes an alternative potential outcome. **b** The values of anisotropy (upper panel) and centripetality (lower panel) were obtained from the RWRC model with different parameters $\alpha$ and $\beta$. The colour maps depict the mean value over 100 independent simulations. Each symbol represents a scenario with different values of $\alpha$ and $\beta$: (△) $\alpha = 1.2$ and $\beta = 2.8$; (◇) $\alpha = 0.8$ and $\beta = 1.6$; (□) $\alpha = 0.2$ and $\beta = 1.2$. **c−e** Maps of population mobility vectors (PMVs) obtained from the simulation results of the above three scenarios. The length and colour of the arrow are proportional to the magnitude of the PMV. Source data are provided as a Source Data file.

paradigms from the vector field, which focuses on validating commuting mobility against Gauss's theorem and defining a scalar potential for identifying mobility basins and contiguous urban areas. In contrast, our research uses the PMV to develop anisotropy and centripetality metrics, offering a quantitative method for analysing overall mobility flow characteristics. This approach represents a development in deciphering urban mobility patterns and contributes valuable insights towards achieving sustainable development goals.

We clustered 60 major Chinese cities into three types using two metrics, anisotropy and centripetality: strong monocentric, weak monocentric, and polycentric cities. Consistent with the existing literature, our findings show that both strong and weak monocentric cities experience an increase in average commuting distance as city size grows[35,36]. This reflects earlier studies indicating that growth in monocentric cities amplifies commuting distances[37]. In contrast, polycentric cities maintain stable commuting patterns, challenging the common view that associates polycentric urban structures with fragmented and unpredictable commuting[38,39]. Our results suggest greater mobility efficiency in polycentric cities, supporting the benefits of polycentric development as advocated by some urban planners[40,41]. This stability contradicts the notion of inefficiency in polycentric cities due to competing centres. Additionally, our study calls for a reevaluation of conventional monocentric and polycentric classifications. While cities are often categorised as one or the other based on

historical and socio-economic factors[42], our nuanced categorisation and observed commuting patterns challenge these rigid classifications, highlighting the dynamic nature of urban forms and evolving commuting behaviours.

Our analysis revealed a consistent increase in anisotropy across all city types at different spatial levels, indicating a more balanced distribution of job opportunities in various directions within urban cores. Polycentric cities showed a less marked increase in anisotropy, reflecting a uniform distribution of job opportunities and spatially dispersed significant employment centres, as opposed to concentration in CBD areas. Conversely, centripetality decreased with increasing spatial levels in all city types, highlighting a core-periphery structure. This decrease was most pronounced in polycentric cities, indicating a reduced tendency for peripheral flows to converge towards the city centre, thus easing centripetal traffic pressures. These patterns, suggesting distinct urban spatial dynamics, warrant further exploration to grasp their implications fully. Moreover, mobile phone data provide valuable spatial insights into human movements, particularly the dynamic properties of mobility spatial directionality. Our research identifies similar spatial directionality characteristics during non-peak hours but significant differences during morning peak hours, indicating unique urban flow patterns at different times of the day.

Unravelling the spatial directionality of urban mobility can offer valuable insights for urban planners and policymakers. For example, if

most commuters travel from the suburbs to the city centre in the morning, public transport and roads can be tailored to manage this flow more effectively. Moreover, tracking directionality trends over time enables planners to forecast future urban growth patterns, which is crucial for long-term planning and sustainable urban development. In essence, mobile phone data provide valuable spatial insights into human movements, particularly the dynamic properties of mobility. The spatial directionality of urban mobility is a vital dimension that, when understood and utilised correctly, can substantially enhance the quality of urban life, promote sustainable development and ensure that cities are more responsive and adaptive to the needs of their residents.

This study lays the groundwork for future enhancements. Although the RWRC model effectively mirrors fundamental patterns, offering crucial theoretical support for understanding the mechanisms influencing spatial directionality, it remains a theoretical construct based on certain assumptions, and it may not fully capture some details of the real world. Thus, this model necessitates validation through empirical studies across diverse contexts. It neglects critical determinants such as traffic congestion, housing prices and socio-economic status, which substantially influence individuals' decisions regarding workplace and residence locations. Integrating these variables could yield models with increased precision and comprehensiveness, significantly advancing our understanding and predictive capabilities regarding urban mobility patterns. Furthermore, our urban centre identification approach solely recognises the optimal centre point, a methodology that has been validated for clarifying urban mobility patterns. Future research could focus on developing algorithms capable of automatically identifying multiple centres in a city, a direction that holds promise as urban areas continue to expand.

## Methods

### Data description
The aggregated and anonymised mobile phone dataset was provided by a Chinese telecommunications operator. The dataset consists of mobile phone signal records collected over a two-month period, representing approximately 90 million residents across 60 cities. The number of anonymised users represents, on average, 18% of the total population, corresponds to the proportion of mobile phone users to the overall population across the 60 cities studied. The operator partitioned all cities into a 0.005° (~0.5 km × 0.5 km) grid cell. The aggregated, non-personally identifiable human movement records represent the number of trips between different grids per hour in August and November 2019 in 60 Chinese cities. Each record contains the date, the hour, the coordinates (longitude and latitude) of the origin grid, the coordinates of the destination grid, and the number of trips. To protect customers' privacy, no individual information or records are available in this dataset. We do not use individual-level data, only anonymized aggregate flows, and this work is exempt from the IRB review of the university according to the Human Research Protection Programme (HRPP).

### Data preprocessing
To obtain the characteristics of human mobility, we processed the dataset from the two dimensions of space and time. In the spatial dimension, we merged the grid into a size of ~1 km × 1 km. In the temporal dimension, from the mobile phone dataset for 25 typical weekdays (Tuesday through Thursday), we calculated the average hourly trips between grids. In addition, human mobility follows the urban rhythm, with high numbers of trips during the day and low numbers during the night. To illustrate our method, we thus selected the time span between 7:00 and 7:59, which corresponds to the typical morning peak hour (Supplementary Note 2 and Supplementary Fig. 2), to quantify the overall characteristics of urban mobility and to classify cities.

### Extracting the urban area and corresponding population
The mobile phone dataset we used for this study was recorded according to the administrative region of each city. However, these administrative regions are not subject to either socioeconomical or morphological factors. To obtain a unified and harmonised delineation of cities, it is necessary to extract urban areas that go beyond the administrative regions defined by the national authorities. Here, we adopted the global human settlement dataset, which defines an urban area as a contiguous area with a density of at least 1500 inhabitants per square kilometre or with most built-up land cover being coincident with a minimum of 50,000 inhabitants[43]. Supplementary Tables 1, 2, Supplementary Note 3 and Supplementary Fig. 3 provide a detailed description of the extracted results. In all our analyses, we concentrate on quantifying the overall characteristics of mobility flows originating from urban areas, rather than those outside urban areas. This is because urban mobility can be more strictly compared between urban areas. On the other hand, regions outside urban areas can be heterogeneous both within and between cities, which makes city-level weighted averaging of their parameters problematic[44].

### Metrics
The OD matrix is critical for analysing urban mobility characteristics, encapsulating detailed information about the flow of people at a given spatial scale and period[11]. It is an $m \times n$ matrix, where $m$ represents the number of origins, $n$ represents the number of destinations, $T_{ij}$ denotes the flow from origin $i$ ($i = 1, 2, ..., m$) to destination $j$ ($j = 1, 2, ..., n$), and $O_i$ denotes the outflow from location $i$, calculated as $O_i = \sum_{j \neq i} T_{ij}$. For a granular analysis, we segment each urban area into the 1 km × 1 km square grid, extracting the OD matrix and grid coordinates. The mobility flow $T_{ij}$, indicating the number of individuals moving from $i$ to $j$, embodies both magnitude and direction. Therefore, we represent $T_{ij}$ as vectors, incorporating spatial directionality into the analysis.

The concept of anisotropy and centripetality metrics can simplify and reduce the dimensionality of a large-scale mobility network, while preserving the spatial directionality information related to human mobility to the greatest extent possible. The PMV is a vector that includes magnitude $\lambda_i$ and direction $\theta_i$. For a given location $i$, the magnitude of the PMV is defined as

$$\lambda_i = |\vec{T}_i| = |\sum_{j \neq i} \frac{T_{ij}}{O_i} \vec{u}_{ij}|, \tag{4}$$

where $\lambda_i$ ranges from 0 to 1. It can be noted that $\lambda_i$ is equal to 1 when all flows $T_{ij}$ originating from location $i$ are oriented in the same direction, and 0 when the flows $T_{ij}$ from location $i$ are spatially symmetrical. Hence, the direction-dependent property $\lambda_i$ reflects the degree of imbalance in the direction distribution of outflow, and we term $\lambda_i$ the anisotropy of outflow from location $i$. The higher the value of $\lambda_i$ (the more concentrated the flows are in a specific direction), the stronger the anisotropy is, and vice versa. After calculating the anisotropy of each location, we computed the anisotropy of urban mobility by taking a weighted average of each location's anisotropy, with the outflow of each location as the weight[44]

$$\Lambda = \frac{\sum_i O_i \lambda_i}{\sum_i O_i}, \tag{5}$$

where $\Lambda$ is the anisotropy of urban mobility, reflecting the overall imbalance degree in the direction distribution of all flows.

For the direction $\theta_i$, we use a common reference point to measure $\theta_i$ consistently across locations. Specifically, we define $\theta_i$ as the relative angle to the city centre $C$, which is also called the allocentric direction[45], that is, the mobility angle in reference to the city centre.

We then calculate the direction $\theta_i$ of the PMV using the law of cosines

$$\theta_i = \arccos \frac{\vec{T}_i \cdot \vec{u_{iC}}}{|\vec{T}_i||\vec{u_{iC}}|}, \quad (6)$$

where $\vec{u_{iC}}$ is the unit vector from $i$ to $C$ and $\theta_i$ ranges from 0 to $\pi$. It can be seen that $\theta_i$ is 0 if $\vec{T}_i$ is oriented toward the city centre and $\pi$ if $\vec{T}_i$ is oriented in the opposite direction. As $\vec{T}_i$ results from aggregating the characteristics of all flows from an origin, the relative direction $\theta_i$ illustrates the overall tendency of flows from a specific location to orient towards the city centre. Consequently, the direction $\theta_i$ signifies the degree of orientation towards the city centre in terms of location mobility, allowing us to define the centripetality of outflow from location $i$ as

$$\gamma_i = 1 - \frac{\theta_i}{\pi}, \quad (7)$$

where $\gamma_i$ ranges from 0 to 1. The higher the value of $\gamma_i$ (the more oriented toward the city centre the flows), the stronger the centripetality is, and vice versa. Similarly, after calculating the centripetality of each location, we compute the centripetality of urban mobility by taking the weighted average of each location's centripetality, with the outflow of each location as the weight[44]

$$\Gamma = \frac{\sum_i O_i \gamma_i}{\sum_i O_i}, \quad (8)$$

where $\Gamma$ is the centripetality of urban mobility, reflecting the overall orientation degree toward the city centre of all flows.

### Identification of the city centre
Here, we develop a data-driven method, called the maximum centripetality algorithm (MCA), to identify the city centre. We define city centre $C$ as the location where the centripetality $\Gamma$ of urban mobility is maximum. The algorithm procedure is as follows.

Step 1: Select an initial centre $i_c$ and calculate the urban mobility centripetality.

Step 2: For each location in the city, repeat Step 1.

Step 3: Among all locations, select the location corresponding to the maximum centripetality of urban mobility as the city centre.

Our algorithm provides an automated and systematic means of identifying the city centre using human mobility data. The results identified with the MCA are shown in Supplementary Table 3.

Our analysis, utilising mobile phone data, identifies a high number of areas of interest (AOIs) near the urban centres (Supplementary Fig. 4), indicating an underlying pattern of land use activity that mobile phone data can unveil. This not only validates the relevance, utility, and robustness of our identified urban centres but also indirectly confirms their accuracy. Furthermore, for cities exhibiting a polycentric model, we have identified multiple city centres, see Supplementary Note 4 and Supplementary Fig. 5.

### Average commuting distance
We utilise the Euclidean distance between the origin and destination coordinates to represent commuting distance. Although Euclidean distance offers a simplified view of spatial relationships, it effectively gauges the proximity between commuting start and end points, residences and workplaces, and it encapsulates the general traits of commuting patterns. For a given city, the average commuting distance during the observation period is

$$D = \frac{\sum_{i,j} T_{ij} d_{ij}}{\sum_{i,j} T_{ij}}, \quad (9)$$

where $i$, $j$ represent locations within the urban area and $d_{ij}$ is the Euclidean distance between $i$ and $j$.

### Simulation
We conduct simulations involving $N$ individuals selecting their workplaces and residences within an $l \times l$ simulation space. Initially, the first simulated individual chooses the central grid ($l/2$, $l/2$) as both workplace and residence. Subsequent individuals select their workplaces according to equation (2) (i.e., $P_j \propto N_j^{\alpha}$) and their residences based on equation (3) (i.e., $Q_{ij} \propto e^{-d_{ij}/\beta}$). The model operates with parameters $N = 10^4$ and $l = 50$. Variations in anisotropy and centripetality are explored through adjustments in $\alpha$ and $\beta$, with numerical results averaged over 100 independent simulations.

### Reporting summary
Further information on research design is available in the Nature Portfolio Reporting Summary linked to this article.

## Data availability
The authors declare that the data supporting the conclusions are described in the paper and the Supplementary Information. The mobile phone data are not publicly accessible because of privacy and contractual concerns. Information about the conditions and limitations of the data can be found in http://www.smartsteps.com/product/dataset.php. Submit your data access request by contacting the sales team via the customer page at http://www.smartsteps.com/about/contactus.php, where they will promptly facilitate your request process. Source data are provided with this paper.

## Code availability
The analysis was conducted using Python. Code to reproduce the main results is available at Code Ocean (https://codeocean.com/capsule/9118584/tree/v1).

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

## Acknowledgements

P.Z. is supported by the National Natural Science Foundation of China (41925003 and 42130402), and the Shenzhen science and technology programme (JCYJ20220818100810024 and KQTD20221101093604016). X.-Y.Y. is supported by the National Natural Science Foundation of China (72271019). We are also grateful for the insightful discussion with Dr. Z. Gong, Dr. S. Jiang and Dr. E. Liu.

## Author contributions

P.Z. and H.W. contributed equally to this work. P.Z., H.W., Q.L. and X.-Y.Y. designed the research. P.Z. and X.-Y.Y. contributed analytic tools. H.W. and X.-Y.Y. performed the research. P.Z., H.W., Q.L. and X.-Y.Y. analysed the data. P.Z., H.W., Q.L., X.-Y.Y. and J.L. wrote the paper.

## Competing interests

The authors declare no competing interests.
