## [Peer Review File · Nature Communications]

Unravelling the spatial directionality of urban mobilityReviewer #1 (Remarks to the Author):

Dear Editor and dear authors,

Thank you for giving me the exciting opportunity to review and study respectively a very interesting manuscript on identifying, understanding and contextualising the patterns and dynamics of urban mobility directionality. This is a very labour-intensive work negotiating an issue, maybe under-represented in the literature currently, with critical interdisciplinary importance since it is underpinning the decision-making for transport and urban development planning and management.

The topic is original, the paper well-written, the method used innovative (pretty impressed with the sample: 90 million mobile phone users across 60 Chinese cities over a 2-month period) and the analysis solid and well-presented. This work clearly advances the state of the art and adds value to the Nature Communications literature.

I have only two suggestions for improvement that are of minor nature and scale. Other than those I do feel this is a publishable work.

First I would have like the authors to end their introduction with a couple of lines describing the remaining paper's structure. This will support the reader in navigating better to what comes next.

Second, I would have liked to see a small commentary at the end of the paper (a concluding remark) that reflects on the possible limitations of the study, notes on what could have been possibly improved and links the work with future research pathways (i.e. what's next for the researchers in this topic?).

Looking forward to seeing the R1 version of the work. Well done.

Prof. Alexandros Nikitas

Reviewer #2 (Remarks to the Author):

Main contribution of this paper lies in accounting for directionality in human mobility patterns, an overlooked but critical aspect of understanding patterns of spatial interactions. Two metrics: 'anisotropy' and 'centripetality' are developed and employed to quantify the directional characteristics based on a large dataset of some 90 million mobile phone users across 60 cities in China.

The paper presents several useful insights and has the potential to add value to the existing literature on unravelling urban spatial structures using human mobility data. That said, the paper is quite hard to follow and understand, mainly because it's written in very technical language. It would help if the technical jargons are minimized in favour of a more accessible, easy to understand language that relates the findings of the analysis back to the real-world. My detailed comments below are intended to help the authors improve clarity and strengthen the paper.

[1] The background literature presented is useful, but I think it ignores important conceptual foundations that would inform the analysis presented in this paper. Ultimately, the paper helps to illustrate various urban spatial structure such as monocentricity and polycentricity, yet the literature review does not really engage with previous work in this field.

[2] The dataset consists of mobile phone signal records collected over a 2-month period: Could you specify the year and exact months? Also, any justification as to the 60 cities represented in the analysis?

[3] "The number of anonymised users represents on average 18% of the total population"—do you mean 18% on the average of each of the 60 cities?

[4] What is PMV? Can you write it in full when you use it for the first time?

[5] "For a granular analysis, we segment each urban area (see Methods, Supplementary Note 2 and Supplementary Fig. 2) into a high-resolution square grid, extracting the OD matrix and grid coordinates". What is this high resolution? Is it different from the 1km x 1km spatial unit you

mentioned earlier?

[6] Could you define these two concepts of 'anisotropy' and 'centripetality' first before going on to specify their respective equations?

[7] "Although the geodesic distance is a simplified representation of reality, it effectively captures the essential aspects of commuting behaviour and urban spatial patterns"—what essential aspects are captured? Could you be specific? Also, is it possible to derive specific routes and hence network-based measurement of distances from the mobile trajectory data? This would provide a more realistic estimation of distances compared to the simple geodesic distance?

[8] In the results, the results of two metrics applied are reported for only three cities out of the 60—Beijing, Tianjin and Foshan. Could you explain why?

[9] "The results indicate distinct patterns in these cities, with differences in anisotropy ($\Lambda_{\text{Beijing}}=0.40$, $\Lambda_{\text{Tianjin}}=0.28$ and $\Lambda_{\text{Foshan}}=0.30$), and centripetality ($\Gamma_{\text{Beijing}}=0.92$, $\Gamma_{\text{Tianjin}}=0.87$ and $\Gamma_{\text{Foshan}}=0.65$), elucidating diverse urban mobility characteristics." What do these values mean in simple terms? I think it would be useful to bring the results back into the real-world by interpreting what they mean. As it stands, the paper is quite Jargony and would require presenting and explaining the results in much simple terms and leaving the technical aspects to the methodology section.

[10] The definition of city centres relies on mobile phone data. Would it have been even more useful and robust combining this data with some kind of point of interest or land use data? I think there is an underlying land use activity pattern that such data would reveal and enrich the analysis presented?

[11] "We explore the link between average commuting distance and city size to understand urban mobility patterns" how do you define size—is it in terms of physical extent or population size? Please clarify. In the subsequent sentences, you use 'urban area' e.g as in "In monocentric cities, there is a strong positive correlation between commuting distance and urban area"—here again, urban area is quite vague. If it's the physical size of the urban area, then please be specific.

[12] "Importantly, they reveal diverse growth trajectories in commuting distances relative to city size, offering novel perspectives on the evolution of urban mobility systems"—I'm not sure if I understand this sentence. What you mean by diverse growth trajectories in commuting distance? I don't see any analysis and results presented on the growth trajectories of commuting distance? Similarly, I do not see any analysis presented on the evolution of urban mobility systems? All the analysis presented is static/snap short of commute flows within a peak hour period so I'm not sure I agree with the claims being made here about 'trajectories' and 'evolution'. Please rewrite.

[13] "To facilitate comparative analysis across varying city sizes, we normalised urban spaces into 5 spatial levels (Fig. 4a), based on the principle of equal trip generation [17, 34]. Here we use $l = 5$ for all cities, while our results are robust against changes of this value (Supplementary Note 6, Fig. 10)." What are these five spatial levels? Can you outline them briefly in the narrative?

[14] Section iii mechanism of anisotropy and centripetality—it's not entirely clear to me what this aspect is seeking to add and most importantly how it relates to the analysis preceding it. My view is that the paper is probably trying to do too much with the addition of this section. In doing so, the simulation is not explained in detail as one would expect—indeed, I don't see a matching methodology in the methods section for the simulation presented here. Perhaps, this needs separating into another paper so that the current papers focuses on the typology of spatial structures derived from the human mobility data?

[15] There are too many supplementary analysis, results figs etc. It would help being selective and presenting the most important results.

Reviewer #3 (Remarks to the Author):

Overall comments

Overall, this is a nice work. Congratulations. The main contribution is in relation to "directionality" in urban mobility towards the "city center". The paper will benefit from a clearer distinction of "directionality" in relation to the non-spatial dimension and the spatial dimension, both conceptually and methodologically.

The introduction and literature review highlight that the objective and main innovation of the paper

is on spatial directionality, which cannot be captured by an origin-destination (OD) matrix (Supplementary Figure 1). Non-spatial directionality, that is, the imbalance of flows within cities, has been very well examined in the literature but spatial directionality is not. The latter, as pointed out by the authors, is a clear research gap.

After reading the paper, I still find that we have not advanced much towards understanding the spatial configuration of flows beyond the OD matrix (Supplementary Figure 1). In this paper, the authors identify a "city center" in each city and then measure the directionality in urban mobility in relation to the "city center". With reference to Figure 1, the method, for example, first identifies A as the city center and then examine all flows in relation to it rather than captures all different flows across A to D in a spatial manner (with spatial orientation), as demonstrated by the arrows in (b) and (c). This meaning of "directionality to city centers" underlies the entire analysis. The subsequent cluster analysis (with the three clusters of "strong monocentric", "weak monocentric", and "polycentric") further reinforces the nature of the methodology and the contribution of the paper. I think this should be clarified in the title and throughout the paper.

Without reference to spatial orientation, this paper uses "directionality" to mean the "direction of flows" being the balance of inflow and outflow. λ essentially measures whether the outflow to a location is dominant in the total outflow from an origin. Here, anisotropy is introduced (below equation 1). Again, studies on non-spatial "directionality" or the imbalance of inflow and outflow are enormous. The introduction of θ in the population mobility vector (PMV) starts to make the analysis spatial and interesting. Yet, it depends on the identification of a "city center" in the first place. I wonder whether we can make θ in relation to the true North. Then, θ can really have universal applicability and be compared for different cities worldwide. The main direction towards the "city center" will still become dominant in the PMV. Additional measures, such as the centripetality in relation to the most dominant "direction" of the "city centre", will still follow. The centripetality measure can keep the existing definition of "city center" as this refers to the gravitational force of the "city center".

Can the next stage be a variable number of "city center" (instead of being fixed as one) based on the mobile phone data? Once the ideal/optimal "city centers" are identified, the centripetality measure can be adjusted in relation to the nearest "city center". The focus on identifying ways of understanding a polycentric urban form is important as cities are getting bigger and bigger. Would this be a promising direction for further research? Please explore.

Specific comments

1) Part I on Introduction

a. With reference to the above, distinguish between the non-spatial and spatial dimensions of "directionality", and emphasize on the contribution towards understanding "directionality" to the dominant city center of each city.

2) Part II on Results

a. Part A, results of three cities, that is, Beijing, Tianjin and Foshan, are suddenly presented without any background. Perhaps, the general results of the highest and lowest values of anisotropy and centripetality (among all 60 cities) should be presented to provide the broad picture.

b. Part B, why is the relative direction still meaningful? For instance, with reference to Figures 1(b) and (c), the relative direction (red arrow) does not reflect the most dominant flow (from green to the blue patch on the top right).

c. Part C is good and well-written.

d. Part D, it is not clear how spatial levels are defined (even after reading Supplementary notes). Please explain more clearly beyond "we normalized urban spaces into l spatial levels ..., based on the principle of equal trip generation". Please elaborate and explain the theoretical significance of the "spatial hierarchical structure". What does $l=5$ suggest in reality? See also my comments on Supplementary Note 6 below.

e. Part E, the main analysis is based on Fig. 5. Yet, its presentation should be improved. For instance, putting the Insert within (c) (with a different scale) is confusing. With the three diagrams, there is perhaps no need for the Insert again. The colours are badly chosen, with late evening and near midnight (2300) as red. Please try having green and blue at the two ends of

early morning and late night (suggesting nighttime) and red and yellow for the midday. Also, "Error bars represent the standard errors" is irrelevant. Can the variations be summarized statistically for comparison?

3) Part III on Mechanism

a. The RWRC model is fine. Is this part purely theoretical? Do you have any idea about the empirical values of α and β across the 60 cities?

4) Part IV on Discussion

a. Generally fine. See overall comments above.

5) Part V on Methods

a. The authors suggested that the dataset consists of mobile phone signal records collected over a 2-month period, representing approximately 90 million residents across 60 cities. Yet, in Part B, it seems that only data of the peak hour (7:00-7:59) of 25 weekdays were used for the cluster analysis. What proportion of the dataset does this represent? Does this only apply to Part B of Section II on Results? Make this clear in the main text.

b. Then, in Part C, it says that actually only the urban areas are considered as "origins" of human mobility flows. Does this apply to all Results reported in Section II? From Fig. 2, it seems that the "urban areas" may vary from about half to one-tenth of the "city boundary". Again, what proportion of the dataset does this represent? Please provide in Supplementary Table 2 the size of the urban area (now listed?), the size of the city (based on city boundary), the share of the former in the latter, and the population of the urban area and city, respectively.

6) Supplementary Figures

a. Figure 1, are the arrows between B and C in diagrams (b) and (c) correct?

b. Figure 5, should (c) be (b) in the title description?

c. Figures 11-13, is there a better way to differentiate the three groups, other than simply describing each of them as "there is a noticeable increase in anisotropy"?

d. Figures 14-16, is there a better way to differentiate the three groups, other than simply describing each of them as "there is a noticeable decrease in centripetality"?

e. Figures 17-19, the diagrams are a bit difficult to read. The x- and y-axes are difficult to see. Also, what do the colours represent? What do the lines represent?

7) Supplementary Notes

a. Supplementary Note 6, this has to be re-written. There are many unclear terms, such as "inhomogeneity" and "equal mass". The main logic of the "spatial hierarchical structure" is also not clear. It needs to be elaborated clearly, instead of just stating that "we model urban space as a hierarchy of l levels according to the principle of equal trip occurrence".

Response to reviewer comments

Reviewer #1 (Remarks to the Author):

Thank you for giving me the exciting opportunity to review and study respectively a very interesting manuscript on identifying, understanding and contextualising the patterns and dynamics of urban mobility directionality. This is a very labour-intensive work negotiating an issue, maybe under-represented in the literature currently, with critical interdisciplinary importance since it is underpinning the decision-making for transport and urban development planning and management. The topic is original, the paper well-written, the method used innovative (pretty impressed with the sample: 90 million mobile phone users across 60 Chinese cities over a 2-month period) and the analysis solid and well-presented. This work clearly advances the state of the art and adds value to the Nature Communications literature.

Response: We are deeply grateful for the reviewer's thoughtful and encouraging feedback on our manuscript. The acknowledgment of our manuscript as an engaging exploration into the intricacies of urban mobility directionality is greatly appreciated. We are particularly thankful for the recognition of the originality of the topic, the clarity of the paper, the innovativeness of our methodology, and the solidity of our analysis, especially given the extensive dataset encompassing 90 million mobile phone users across 60 Chinese cities. Your affirmation that our work advances the state of the art and contributes valuable insights to the Nature Communications literature is both motivating and affirming. It is our sincere hope that our findings will indeed foster meaningful advancements in the fields of transport and urban development planning and management. Thank you once again for your insightful assessment and for highlighting the interdisciplinary importance of our research.

I have only two suggestions for improvement that are of minor nature and scale. Other than those I do feel this is a publishable work.

First I would have like the authors to end their introduction with a couple of lines describing the remaining paper's structure. This will support the reader in navigating better to what comes next.

Response: We thank the reviewer for this constructive suggestion. Acknowledging the importance of guiding our readers through the manuscript, we have taken the initiative to revise the introduction (pp.2-3), incorporating a concise overview of the paper's structure at its discussion (p.3). This addition is intended to enhance the reader's navigation and understanding of the subsequent sections. We are confident that this improvement will make the manuscript more reader-friendly and facilitate a smoother transition between sections. Again, thank you for this valuable recommendation, which we believe has significantly enriched our manuscript's clarity and accessibility.

Second, I would have liked to see a small commentary at the end of the paper (a concluding remark) that reflects on the possible limitations of the study, notes on what could have been possibly improved and links the work with future research pathways (i.e. what's next for the researchers in this topic?).

Response: We deeply appreciate your recommendation to discuss the study's limitations and future research directions. Acknowledging this valuable guidance, we have refined our manuscript to include reflections on potential areas for improvement and the broader implications for future research (p.11). This entails an acknowledgment of the simplified assumptions in our theoretical model and their implications for capturing the complexities of urban mobility. Furthermore, we highlight the reliance on Euclidean distances for estimating commuting distances as a methodological limitation, suggesting that advancements in data collection could offer more nuanced insights into urban mobility patterns. By incorporating these considerations, we aim to not only enhance the integrity of our study but also to underscore the importance of continued exploration in this field. Your suggestion has been instrumental in enriching our manuscript's contribution to the discourse on urban mobility.

Reviewer #2 (Remarks to the Author):

Main contribution of this paper lies in accounting for directionality in human mobility patterns, an overlooked but critical aspect of understanding patterns of spatial interactions. Two metrics: 'anisotropy' and 'centripetality' are developed and employed to quantify the directional characteristics based on a large dataset of some 90 million mobile phone users across 60 cities in China.

The paper presents several useful insights and has the potential to add value to the existing literature on unravelling urban spatial structures using human mobility data. That said, the paper is quite hard to follow and understand, mainly because it's written in very technical language. It would help if the technical jargons are minimized in favour of a more accessible, easy to understand language that relates the findings of the analysis back to the real-world. My detailed comments below are intended to help the authors improve clarity and strengthen the paper.

Response: We sincerely appreciate the reviewer's thoughtful feedback and recognition of the potential contribution our manuscript offers to the literature on urban spatial structures and human mobility data. We acknowledge the concern regarding the manuscript's technical density and the challenge it may pose to readability. In response to your invaluable suggestions, we have endeavoured to refine our manuscript, consciously reducing the use of technical jargon and aiming for a more accessible narrative (pp.2-4). Our revisions seek to bridge the gap between complex analytical findings and their practical implications in the real world, ensuring that the insights are clear and relevant to a broader audience. This effort to enhance clarity and readability is guided by your detailed comments, for which we are profoundly thankful. We believe these adjustments significantly strengthen the manuscript and enhance its contribution to the field.

[1] The background literature presented is useful, but I think it ignores important conceptual foundations that would inform the analysis presented in this paper. Ultimately, the paper helps to illustrate various urban spatial structure such as monocentricity and polycentricity, yet the literature review does not really engage with previous work in this field.

Response: We deeply appreciate the reviewer's insightful critique regarding the literature review

section of our manuscript. We acknowledge the omission of critical conceptual foundations, such as monocentricity and polycentricity, which are essential for grounding our analysis in the broader discourse on urban spatial structures. In response to your valuable feedback, we have thoroughly revised the literature review to include a more comprehensive discussion of these foundational concepts. We now explicitly address how our work, with its focus on spatial directionality of urban mobility, complements and diverges from traditional studies of urban spatial structures that emphasise population density, infrastructure, and employment distribution (pp.2,14). This enhancement aims to situate our analysis within a well-established theoretical framework, offering readers a clearer understanding of our contributions to the field and the novelty of our insights into urban spatial configurations. We are grateful for the opportunity to refine our manuscript based on your recommendations, and we believe these revisions significantly strengthen the paper's theoretical grounding and relevance to existing literature.

[2] The dataset consists of mobile phone signal records collected over a 2-month period: Could you specify the year and exact months? Also, any justification as to the 60 cities represented in the analysis?

Response: We appreciate your valuable query regarding the specifics of our dataset and the selection rationale for the 60 cities included in our analysis. In response to your suggestion, we have made the timeframe of our dataset more explicit within the manuscript, clarifying that the mobile phone signal records were collected in August and November 2019 (p.3, 12). To address your query on the selection of cities, these were primarily chosen to include China's directly-administered municipalities and provincial capitals, which are indicative of higher economic development levels. This selection was made to ensure that our analysis reflects the mobility patterns of China's most representative major cities, offering insights into the broader urban mobility dynamics across varied socioeconomic contexts (p.3). The data provider specifically supplied information for these 60 cities, facilitating a comprehensive analysis of urban mobility across significant metropolitan areas in China. We have updated our manuscript accordingly to incorporate this clarification both in the data description and the introduction, aiming for greater transparency and to aid readers in understanding the scope and basis of our research. We are grateful for the opportunity to enhance our manuscript with these details and hope this addresses your concerns adequately.

[3] "The number of anonymised users represents on average 18% of the total population"—do you mean 18% on the average of each of the 60 cities?

Response: We are grateful for the reviewer's attentive observation and query regarding our mention of the 18% anonymised users. To clarify, indeed, the figure of 18% represents the average proportion of mobile phone users to the total population, considered collectively across the 60 cities included in our study. In light of your valuable feedback, we have updated the manuscript to clearly convey this information, ensuring there is no ambiguity in our description. The revised sentence now accurately reflects that the anonymised user data corresponds to, on average, 18% of the overall population when looking at the collective demographic of the 60 cities (p.12). We appreciate your keen attention to detail, which has helped enhance the clarity and precision of our manuscript. Thank you for enabling us to improve our communication of these important data characteristics.

[4] What is PMV? Can you write it in full when you use it for the first time?

Response: We deeply appreciate your attention to the clarification of the acronym PMV. Following your valuable feedback, we have ensured that the term “population mobility vector (PMV)” is fully defined at its initial mention in the introduction (p.3). This addition serves to introduce the concept clearly and sets the stage for discussing its role in developing the two pivotal metrics: anisotropy and centripetality. We are grateful for your insight, which has significantly contributed to enhancing the readability and comprehensibility of our manuscript. Thank you for guiding us to make this necessary improvement.

[5] “For a granular analysis, we segment each urban area (see Methods, Supplementary Note 2 and Supplementary Fig. 2) into a high-resolution square grid, extracting the OD matrix and grid coordinates”. What is this high resolution? Is it different from the 1km x 1km spatial unit you mentioned earlier?

Response: We express our gratitude to the reviewer for highlighting the ambiguity concerning the term “high-resolution square grid.” Upon reflection, we recognise the need for clearer expression in our manuscript. The “high-resolution square grid” indeed refers to the 1km*1km spatial unit mentioned earlier in our text (p.12). To eliminate any confusion, we have now explicitly stated this in the manuscript, specifying that for granular analysis, each urban area is segmented into a 1km*1km square grid, from which we extract the OD matrix and grid coordinates.

[6] Could you define these two concepts of ‘anisotropy’ and ‘centripetality’ first before going on to specify their respective equations?

Response: We gratefully acknowledge your constructive suggestion to define the concepts of “anisotropy” and “centripetality” before introducing their corresponding equations. In response to your feedback, we have revised the manuscript to incorporate definitions of these two critical metrics at the outset. These revisions clarify that anisotropy quantifies the uneven distribution of urban mobility across different directions, with higher values indicating a concentration of flows in specific directions. Centripetality, on the other hand, measures the orientation of flows towards the city centre, where higher values suggest a dominant movement towards this central point (p.4). Together, these metrics offer insights into the spatial directional characteristics of mobility, aiding in distinguishing between monocentric and polycentric urban mobility patterns. Furthermore, as per your recommendation, we have relocated the detailed mathematical formulations to the appendix, ensuring the main text remains accessible while still providing the necessary depth for interested readers.

[7] “Although the geodesic distance is a simplified representation of reality, it effectively captures the essential aspects of commuting behaviour and urban spatial patterns”—what essential aspects are captured? Could you be specific? Also, is it possible to derive specific routes and hence network-based measurement of distances from the mobile trajectory data? This would provide a more realistic estimation of distances compared to the simple geodesic distance?

Response: We appreciate your insightful inquiry regarding the specific aspects of commuting behaviour and urban spatial patterns captured by the geodesic distance, and the potential for deriving more accurate network-based measurements from mobile trajectory data. Upon reflection, we realise that our initial reference to “geodesic distance” may have caused confusion, and we have corrected this terminology to “Euclidean distance” in our manuscript to better convey our intent (p.13).

The “essential aspects” we aimed to highlight pertain to the Euclidean distance’s utility in providing a straightforward measure of proximity between origins and destinations, such as homes and workplaces. This distance serves as an initial gauge for assessing the spatial dimension of commuting patterns. However, based on your feedback, we acknowledge that Euclidean distance, while offering a baseline understanding, may not fully capture the intricacies of urban commuting behaviours and spatial relationships (p.13).

In light of this, we have amended our manuscript to clarify: “We utilise the Euclidean distance between the origin and destination coordinates to represent commuting distance. Although Euclidean distance offers a simplified view of spatial relationships, it effectively gauges the proximity between commuting start and end points, residences and workplaces, and encapsulates the general traits of commuting patterns”, thereby offering a basic framework to understand commuting patterns.” (p.13)

We regret to inform that our dataset comprises only grid-based flow data, lacking individual trajectory data, which precludes us from deriving specific routes and calculating network-based distances from mobile phone data. Although theoretically possible, modelling these flows onto a transport network for more accurate distance measurements introduces a level of complexity and computational intensity that far surpasses Euclidean distance calculations. Given the simplicity and reduced computational demands of Euclidean distances, this method is favoured for large-scale urban studies where efficiency and straightforward analysis are crucial. This rationale underpins our choice, as highlighted in our manuscript, to utilise Euclidean distance as an effective, albeit simplified, measure for understanding urban commuting patterns and spatial relationships (p.13). Your insights have been invaluable in refining our discussion on this matter (p.11), and we are grateful for the opportunity to clarify our methodological approach. Your suggestion to explore network-based measurements for a more realistic estimation of distances is indeed valuable. While our current dataset and analysis framework rely on Euclidean distances, we recognise the potential benefits of incorporating network-based distances in future research for a more nuanced understanding of urban mobility.

[8] In the results, the results of two metrics applied are reported for only three cities out of the 60—Beijing, Tianjin and Foshan. Could you explain why?

Response: We deeply appreciate your inquiry regarding the selective presentation of results for only three out of the sixty cities analysed in our study—specifically Beijing, Tianjin, and Foshan. Our rationale for this focused approach was to provide a clear, real-world application of the concepts of anisotropy and centripetality within diverse urban contexts, thus facilitating a more intuitive understanding of these metrics for our readers. The selected cities exhibit distinct differences in their mobility patterns, serving as illustrative examples that enrich the discussion of our findings.

Beijing, with high anisotropy and centripetality values ($\Lambda=0.40$, $\Gamma=0.92$), epitomises a

monocentric urban mobility pattern. In contrast, Foshan demonstrates lower values in both metrics ($\Lambda=0.30$, $\Gamma=0.65$), indicative of a polycentric pattern, whereas Tianjin, with lower anisotropy but higher centripetality ($\Lambda=0.28$, $\Gamma=0.87$), represents a nuanced, possibly transitional, pattern between monocentric and polycentric structures (p.4).

This deliberate selection aims to cover a spectrum of urban structures and mobility dynamics, providing a broad overview of how varying anisotropy and centripetality values manifest in real-world scenarios. We believe this approach significantly aids in conceptualising and understanding the implications of our metrics on urban mobility patterns. Your feedback has been invaluable in clarifying this aspect of our methodology, and we are grateful for the opportunity to elucidate our rationale behind the choice of cities presented in our results.

[9] “The results indicate distinct patterns in these cities, with differences in anisotropy ($\Lambda_{\text{Beijing}}=0.40$, $\Lambda_{\text{Tianjin}}=0.28$ and $\Lambda_{\text{Foshan}}=0.30$), and centripetality ($\Gamma_{\text{Beijing}}=0.92$, $\Gamma_{\text{Tianjin}}=0.87$ and $\Gamma_{\text{Foshan}}=0.65$), elucidating diverse urban mobility characteristics.” What do these values mean in simple terms? I think it would be useful to bring the results back into the real-world by interpreting what they mean. As it stands, the paper is quite Jargony and would require presenting and explaining the results in much simple terms and leaving the technical aspects to the methodology section.

Response: We are immensely grateful for the reviewer’s suggestion to clarify the significance of our findings in more accessible terms. Recognising the technical nature of our initial presentation, we have endeavoured to contextualise the values of anisotropy (Λ) and centripetality (Γ) within the tangible urban experience.

In simpler terms, higher anisotropy (Λ) values, like Beijing’s 0.40, signify a concentrated distribution of travel flows in specific directions, typical of cities with a focused urban core or monocentric pattern. Conversely, lower anisotropy, as seen in Tianjin ($\Lambda=0.28$) and Foshan ($\Lambda=0.30$), suggests a more dispersed mobility pattern, potentially indicating a move towards polycentricity. Centripetality (Γ) values, such as Beijing’s 0.92, indicate a strong tendency for mobility flows to converge towards the city centre, reinforcing the monocentric model. Lower values, like Foshan’s 0.65, imply a less centralised movement, characteristic of polycentric urban forms. Tianjin’s centripetality ($\Gamma=0.87$), while high, coupled with its lower anisotropy, may suggest a nuanced, transitional urban mobility pattern.

We have revised our manuscript to better illustrate these concepts, aiming to bridge the gap between our technical analysis and its real-world implications. This revision highlights how differing anisotropy and centripetality values provide insights into the city’s underlying mobility structure—whether it leans towards a centralised (monocentric) or decentralised (polycentric) pattern (p.4). Our refined discussion now offers a more intuitive understanding of urban mobility’s spatial characteristics, aligning technical findings with observable urban dynamics.

Your comment has been instrumental in enhancing our manuscript’s accessibility and relevance. We sincerely thank you for prompting this important clarification.

[10] The definition of city centres relies on mobile phone data. Would it have been even more useful and robust combining this data with some kind of point of interest or land use data? I think there is an underlying land use activity pattern that such data would reveal and enrich the analysis presented?

Response: We are grateful for your insightful suggestion to enhance the robustness of our city centre identification methodology by integrating mobile phone data with additional datasets, such as POI or land use data. We wholeheartedly agree that such an approach could unveil underlying land use activity patterns, providing a richer context to our analysis.

In response to your valuable recommendation, we have already taken steps to incorporate Area of Interest (AOI) data into our analysis, as demonstrated in Supplementary Figure 4 (SI p.4). This integration serves to validate the city centres identified through mobile phone data by showing a significant concentration of AOIs in these areas. This convergence not only bolsters the reliability of our identification method but also offers deeper insights into the relationship between urban mobility patterns and land use dynamics (SI p.19).

The integration of AOI data reveals distinct land use activities surrounding the identified city centres, thereby enriching our understanding of urban spatial structures and affirming the accuracy of our methodology. This multidimensional approach underscores the potential of combining diverse data sources to uncover the complex interplay of factors shaping urban environments.

[11] “We explore the link between average commuting distance and city size to understand urban mobility patterns” how do you define size—is it in terms of physical extent or population size? Please clarify. In the subsequent sentences, you use ‘urban area’ e.g as in “In monocentric cities, there is a strong positive correlation between commuting distance and urban area”—here again, urban area is quite vague. If it’s the physical size of the urban area, then please be specific.

Response: We appreciate the reviewer’s attention to detail and understand the importance of clarity in our definitions. We apologise for any confusion caused by our use of the term “city size.” To clarify, our study employs two distinct measures to define city size, enhancing the robustness of our exploration into the link between average commuting distance and city size.

Firstly, we define city size in terms of the physical extent, specifically referred to as “urban area size” in our revised manuscript to avoid ambiguity. This measure pertains to the geographical boundaries of the urbanised area within a city. Secondly, we also consider “urban population size” as an alternative measure of city size, reflecting the number of inhabitants within these urbanised areas (pp.5-6).

By employing both urban area size and urban population size, we aim to provide a comprehensive analysis of how different dimensions of city size relate to urban mobility patterns, particularly average commuting distances. This dual approach allows us to assess the impact of both the physical layout and the demographic density of cities on commuting behaviours.

We believe these clarifications will enhance the reader’s understanding of our findings and the methodological approach we have undertaken. Thank you for guiding us to improve the precision and clarity of our study.

[12] “Importantly, they reveal diverse growth trajectories in commuting distances relative to city size, offering novel perspectives on the evolution of urban mobility systems”—I’m not sure if I understand this sentence. What you mean by diverse growth trajectories in commuting distance? I don’t see any analysis and results presented on the growth trajectories of commuting distance? Similarly, I do not see any analysis presented on the evolution of urban mobility systems? All the

analysis presented is static/snap short of commute flows within a peak hour period so I'm not sure I agree with the claims being made here about 'trajectories' and 'evolution'. Please rewrite.

Response: We are grateful for the reviewer's constructive feedback and apologise for any confusion caused by our previous use of the terms "trajectories" and "evolution". Upon reflection, we acknowledge that our initial discussion may have inadvertently suggested a longitudinal analysis which was not presented in the manuscript. The intent was to illustrate how the characteristics of urban mobility, specifically commuting distances, vary with city size at a given point in time, rather than over a historical timeline.

To address this, we have revised our manuscript to more accurately reflect a cross-sectional analysis of commuting distances across cities of varying sizes, rather than implying a temporal evolution. Our revised discussion now emphasises the observed correlation between commuting distances and city size, highlighting the distinct patterns exhibited by monocentric and polycentric cities. Specifically, in monocentric cities, a strong positive correlation exists between commuting distance and city size, suggesting longer commutes in larger cities. In contrast, polycentric cities show a more stable commuting distance regardless of city size, illustrating diverse patterns in how urban areas are navigated (pp.5-6).

We have also clarified that our references to the "evolution" of urban mobility systems were intended to describe variations in commuting distances as a function of city size and mobility patterns, rather than temporal changes (p.6). We believe these revisions provide a clearer and more accurate depiction of our findings, avoiding any implication of a longitudinal study.

[13] "To facilitate comparative analysis across varying city sizes, we normalised urban spaces into l spatial levels (Fig. 4a), based on the principle of equal trip generation [17, 34]. Here we use $l = 5$ for all cities, while our results are robust against changes of this value (Supplementary Note 6, Fig. 10)." What are these five spatial levels? Can you outline them briefly in the narrative?

Response: We appreciate your inquiry regarding the five spatial levels used in our comparative analysis across cities of varying sizes. The concept of spatial levels is integral to our methodology, aimed at standardising the urban space segmentation for a consistent analytical framework.

In response to rapid urban population concentration, many Chinese cities have evolved into multi-ring road networks, typically around five rings, to accommodate high-density population distribution and increasing transport demands. To transcend the specific nomenclature and structural differences of city rings and ensure analytical consistency across different cities, we opted not to use the existing urban rings as a basis for our spatial segmentation. Instead, we devised a uniform division of "ring zones" based on the principle of equal trip generation. This approach allows for a direct comparison across cities, sidestepping the variability introduced by the unique configurations of individual urban environments (pp.6, SI p.21).

For our study, we defined five spatial levels within each city, segmenting the urban area into concentric zones that each account for an equal share of the city's total trip generation (p.8). This methodological choice is not directly tied to the physical ring roads but is an analytical construct to facilitate our examination of urban mobility patterns in a standardised manner.

We hope this clarification provides a better understanding of the rationale behind our segmentation approach and the significance of the five spatial levels in our analysis. Thank you for

the opportunity to elucidate this aspect of our study.

[14] Section iii mechanism of anisotropy and centripetality—it's not entirely clear to me what this aspect is seeking to add and most importantly how it relates to the analysis preceding it. My view is that the paper is probably trying to do too much with the addition of this section. In doing so, the simulation is not explained in detail as one would expect—indeed, I don't see a matching methodology in the methods section for the simulation presented here. Perhaps, this needs separating into another paper so that the current papers focuses on the typology of spatial structures derived from the human mobility data?

Response: We appreciate the reviewer's insightful feedback on Section iii, regarding the mechanism of anisotropy and centripetality, and its relevance to the preceding analysis. We understand the concern that the addition of this section may seem to extend beyond the scope of the paper's primary focus on urban mobility patterns derived from human mobility data.

The intent behind this section was to delve into the potential underlying causes of the three distinct mobility patterns we identified, aiming to bridge the observation of these phenomena with an understanding of their operational mechanisms—a common approach in complex systems research (p.9). By introducing mechanisms such as employment attraction strength and commuting distance scale, we sought to reproduce these mobility patterns, thereby offering theoretical support for policymakers. This, in turn, is aimed at aiding the design of targeted interventions to guide and optimise the development of urban commuting structures (pp.9-10).

Acknowledging your suggestion for clarity and focus, we have revised the manuscript to ensure a seamless integration of this discussion into the broader narrative. We have also enriched the methods section with a detailed description of the simulation methodology, hoping to address the need for a matching methodology for the simulations presented.

[15] There are too many supplementary analysis, results figs etc. It would help being selective and presenting the most important results.

Response: We are thankful for your recommendation regarding the volume of supplementary analyses, figures, and results included in our submission. In response to your valuable feedback, we have carefully reevaluated the supplementary materials to ensure that only the most critical and directly relevant items are presented. This refinement process aimed to streamline the supplementary content, highlighting those elements that most significantly bolster the main findings and narrative of our paper.

We believe this focused approach enhances the clarity and impact of our supplementary materials, allowing readers to more easily digest and appreciate the core contributions of our research.

Reviewer #3 (Remarks to the Author):

Overall comments

Overall, this is a nice work. Congratulations. The main contribution is in relation to “directionality” in urban mobility towards the “city center”. The paper will benefit from a clearer distinction of

“directionality” in relation to the non-spatial dimension and the spatial dimension, both conceptually and methodologically.

Response: We are grateful for your positive appraisal of our work and your insightful recommendation concerning the clarification of “directionality” in urban mobility towards the “city centre”. In response to your invaluable feedback, we have undertaken a thorough revision to delineate more clearly between the non-spatial and spatial dimensions of directionality, both in conceptual and methodological terms.

This enhancement, now articulated in the abstract, aims to provide a more nuanced understanding of our main contribution and to ensure that readers can fully grasp the significance of directionality in shaping urban mobility patterns (pp.2-3). We believe that this distinction will significantly enrich the paper’s contribution to the field, offering a clearer framework for readers to appreciate the complexities of urban mobility.

The introduction and literature review highlight that the objective and main innovation of the paper is on spatial directionality, which cannot be captured by an origin-destination (OD) matrix (Supplementary Figure 1). Non-spatial directionality, that is, the imbalance of flows within cities, has been very well examined in the literature but spatial directionality is not. The latter, as pointed out by the authors, is a clear research gap.

After reading the paper, I still find that we have not advanced much towards understanding the spatial configuration of flows beyond the OD matrix (Supplementary Figure 1). In this paper, the authors identify a “city center” in each city and then measure the directionality in urban mobility in relation to the “city center”. With reference to Figure 1, the method, for example, first identifies A as the city center and then examine all flows in relation to it rather than captures all different flows across A to D in a spatial manner (with spatial orientation), as demonstrated by the arrows in (b) and (c). This meaning of “directionality to city centers” underlies the entire analysis. The subsequent cluster analysis (with the three clusters of “strong monocentric”, “weak monocentric”, and “polycentric”) further reinforces the nature of the methodology and the contribution of the paper. I think this should be clarified in the title and throughout the paper.

Response: We are appreciative of your constructive feedback regarding the focus on spatial directionality in our paper and the need for clearer communication on how this concept diverges from traditional OD matrix analyses. In light of your comments, we have refined our approach to more explicitly articulate the distinction between non-spatial directionality—represented by the imbalance of flows within cities—and spatial directionality, which captures the nuanced orientation of these flows in relation to the city centre (pp.2-4).

Our methodology, as illustrated in Figure 2, begins with an initial identification of a “city centre” and then progresses to a detailed examination of urban mobility flows in reference to this centre. This process is distinct in its emphasis on the spatial orientation of flows, a perspective that is not adequately captured by the conventional OD matrix. To underscore this methodological shift, we have introduced the concept of the Population Mobility Vector (PMV), calculated through vector summation, which operates independently of the city centre in its initial formulation. However, for a nuanced analysis of PMV in a two-dimensional space, we delve into its magnitude and direction

relative to the city centre, thus linking PMV directly with urban mobility patterns (pp.4-5).

In response to your suggestion, we have considered adding “spatial” before “directionality” in the title to more accurately reflect the paper’s emphasis on the orientation towards the city centre and the distributional imbalances in direction. Such revisions have been made throughout the paper to ensure that the distinction between spatial and non-spatial directionality is consistently clear and emphasised.

We believe these modifications significantly enhance the paper’s clarity and effectively convey our contribution to bridging the identified research gap in spatial directionality. Thank you for guiding us toward a more precise and illuminating presentation of our work.

Without reference to spatial orientation, this paper uses “directionality” to mean the “direction of flows” being the balance of inflow and outflow. λ essentially measures whether the outflow to a location is dominant in the total outflow from an origin. Here, anisotropy is introduced (below equation 1). Again, studies on non-spatial “directionality” or the imbalance of inflow and outflow are enormous. The introduction of θ in the population mobility vector (PMV) starts to make the analysis spatial and interesting. Yet, it depends on the identification of a “city center” in the first place. I wonder whether we can make θ in relation to the true North. Then, θ can really have universal applicability and be compared for different cities worldwide. The main direction towards the “city center” will still become dominant in the PMV. Additional measures, such as the centripetality in relation to the most dominant “direction” of the “city centre”, will still follow. The centripetality measure can keep the existing definition of “city center” as this refers to the gravitational force of the “city center”.

Response: We are truly grateful for your constructive and insightful feedback, especially your suggestion regarding the orientation of the PMV in relation to true North. This perspective indeed offers an intriguing avenue for enhancing the universal applicability of our findings, allowing for comparisons across different cities without reliance on a specific city centre.

Upon careful consideration, we recognise the merit of orienting PMV relative to geographical directions, such as North, which could provide a consistent reference across various urban contexts. However, our study primarily aims to elucidate urban mobility patterns in relation to city centres, which are often hubs of employment opportunities and economic activity. These centres’ locations within a city’s geography—be it coastal, mountainous, or otherwise—influence the general direction of mobility flows, making the relative orientation towards the city centre highly relevant for understanding local commuting behaviours.

For instance, if a city’s CBD is located in the southwest, it’s plausible to observe morning rush hour PMVs predominantly pointing southwest, whilst in another city with a southeastern CBD, PMVs would lean southeast. Such directional tendencies offer valuable insights into the overall flow patterns within each city, aiding in traffic planning and urban development.

Nevertheless, your suggestion prompts us to reflect on the potential for integrating a more geographically oriented approach in future research, which could complement the city-centric analysis presented here. While this study emphasises the angle between PMV and the city centre to gauge the tendency of movements towards or away from the urban core, we acknowledge the significance of considering universal spatial orientations to broaden the applicability of our work.

Can the next stage be a variable number of “city center” (instead of being fixed as one) based on the mobile phone data? Once the ideal/optimal “city centers” are identified, the centripetality measure can be adjusted in relation to the nearest “city center”. The focus on identifying ways of understanding a polycentric urban form is important as cities are getting bigger and bigger. Would this be a promising direction for further research? Please explore.

Response: We are grateful for your constructive feedback and the intriguing suggestion to explore the identification of multiple “city centres” based on mobile phone data, recognising the increasingly polycentric nature of urban forms as cities expand. Your recommendation aligns with a promising direction for future research, underscoring the need to adapt our analytical frameworks to better reflect the complexity of modern urban environments.

In response to your suggestion, we have initiated a preliminary exploration into the concept of polycentric cities and the potential for identifying multiple urban centres. Our current approach, as detailed in Supplementary Figure 5, employs a qualitative methodology to discern cities exhibiting polycentric mobility patterns, with each location color-coded by its centripetality measure (Γ) when regarded as a potential city centre (SI p.5). This analysis has revealed the existence of multiple significant urban cores within these cities, suggesting the presence of both primary and secondary centers of urban activity. This observation challenges the traditional monocentric model and offers valuable perspectives for urban planning and the development of sustainable urban environments (SI p.19).

However, we acknowledge the limitations of our current qualitative approach in identifying these multiple centres and recognise the potential for developing algorithms that can automatically detect multiple urban cores. We intend to explore this possibility in future research, as outlined in our discussion section (p.11).

Specific comments

1) Part I on Introduction

a. With reference to the above, distinguish between the non-spatial and spatial dimensions of “directionality”, and emphasize on the contribution towards understanding “directionality” to the dominant city center of each city.

Response: We thank you for your constructive feedback, prompting us to clarify the distinction between non-spatial and spatial dimensions of “directionality” and their contributions to understanding urban mobility, particularly towards the dominant city centre. Following your valuable suggestion, we have refined our introduction to underscore this differentiation more clearly (pp.2-3).

In recent years, the exploration of human mobility has increasingly employed quantitative metrics to capture the essence of urban travel. Whilst these metrics have traditionally hinged on non-spatial data, such as OD matrices, to derive insights into urban mobility patterns, they offer limited perspective on the spatial orientation of these flows—especially concerning the city centre, which is often a focal point of urban mobility.

To address this gap, our study introduces an analysis of spatial directionality, which considers the directional orientation of mobility flows towards or away from significant urban landmarks, notably the city centre. This approach recognises that urban mobility cannot be fully understood

without considering the spatial context in which it occurs. For example, if the city's CBD were to move to the city's edge, a non-spatial OD matrix would not capture the resultant shift in mobility patterns. However, by incorporating spatial directionality, we can detect significant changes in how residents navigate the urban environment, with mobility patterns realigning towards the new CBD location. This shift, invisible to non-spatial analyses, highlights the critical need for spatial directionality metrics that capture the dynamic nature of urban mobility in relation to the city's structural focal points.

We have made the necessary adjustments to the abstract and introduction sections of our manuscript to highlight these points. By doing so, we aim to provide a clearer and more comprehensive understanding of how our work contributes to bridging the existing gap in the literature on urban mobility, particularly regarding the spatial dimension of directionality towards the city centre.

2) Part II on Results

a. Part A, results of three cities, that is, Beijing, Tianjin and Foshan, are suddenly presented without any background. Perhaps, the general results of the highest and lowest values of anisotropy and centripetality (among all 60 cities) should be presented to provide the broad picture.

Response: We appreciate your insightful suggestion to provide a more comprehensive context before delving into the specific results of Beijing, Tianjin, and Foshan. Understanding the importance of setting a broad picture for our readers, we have revised the presentation of our results to include a preliminary overview of the highest and lowest values of anisotropy and centripetality observed across all 60 cities studied (p.4). This adjustment aims to establish a clearer context and enhance the comprehension of the specific examples presented subsequently. Following your guidance, our discussion now begins with a general analysis that highlights the range of anisotropy and centripetality values encountered in our study, setting the stage for a more detailed examination of the distinct urban mobility patterns in Beijing, Tianjin, and Foshan. This approach allows us to demonstrate the variability and complexity of urban mobility across different cities, thereby providing a clearer understanding of the specific examples that follow. By incorporating this broader perspective, we aim to strengthen the narrative flow of our results section and offer readers a more nuanced appreciation of the diversity in urban mobility patterns we have identified. Your recommendation has been invaluable in enhancing the clarity and impact of our results presentation. Many thanks again for assisting us in improving the quality of our manuscript.

b. Part B, why is the relative direction still meaningful? For instance, with reference to Figures 1(b) and (c), the relative direction (red arrow) does not reflect the most dominant flow (from green to the blue patch on the top right).

Response: We sincerely thank you for your insightful observation regarding the relevance of the relative direction as represented by the red arrow in Figures 1(b) and (c), and its relationship to the most dominant flow. We understand the concern that the red arrow does not directly indicate the most dominant flow from green to the blue patch, and appreciate the opportunity to clarify the significance of this relative direction in our analysis.

The relative direction, as depicted by the red arrow, is indeed the result of aggregating

characteristics of all flows originating from a specific point, rather than highlighting a singular, dominant flow. This aggregation reflects a broader understanding of urban mobility, encapsulating the collective orientation of all flows towards the city centre and offering a comprehensive view of mobility patterns from the specified location. Our aim is to illuminate the integrated mobility characteristic of each point of origin, capturing the essence of all movement flows, not just the most pronounced ones (p.4).

To address your concerns, we have revised our manuscript to more explicitly convey the methodology and reasoning behind the use of relative direction (θ_i) in our analysis (p.13). These revisions aim to underscore the comprehensive nature of the PMV and the relative direction's role in reflecting the overall mobility tendencies within the urban fabric, beyond just identifying the most dominant flow. We hope this clarification enhances the understanding of our approach and the value of considering aggregate flow characteristics in urban mobility studies.

c. Part C is good and well-written.

Response: Many thanks.

d. Part D, it is not clear how spatial levels are defined (even after reading Supplementary notes). Please explain more clearly beyond “we normalized urban spaces into l spatial levels ..., based on the principle of equal trip generation”. Please elaborate and explain the theoretical significance of the “spatial hierarchical structure”. What does $l=5$ suggest in reality? See also my comments on Supplementary Note 6 below.

Response: We sincerely appreciate your request for further clarification regarding the definition of spatial levels and the theoretical significance of the “spatial hierarchical structure.” Recognising the need for a clearer exposition, we have refined our explanation and elaborated on the underlying principles guiding our methodology.

To facilitate a uniform analysis across cities of various sizes, we have normalised urban spaces into hierarchical levels, denoted as l . This normalisation involves segmenting the urban area into l levels, where each level is defined by an equal amount of trip generation. This ensures that the comparative analysis is not biased by the inherent size differences among cities. We have added detailed descriptions and visual aids in the manuscript and supplementary materials to better illustrate how these levels are determined and their relevance to our study (pp.6,8, SI p.21).

The segmentation into spatial hierarchical levels allows us to examine urban mobility with a new lens, highlighting the spatial directionality of movements within cities. This mesoscale approach is pivotal for our understanding of urban mobility patterns, as it reveals the nuanced differences in how mobility is distributed across an urban area. By quantifying spatial directionality at different hierarchical levels, we can uncover insights into the organisation and flow of urban spaces, providing valuable information for urban planning and management strategies aimed at enhancing city liveability and efficiency (p.8).

The choice of $l=5$ levels is inspired by the common structural configuration of many cities, particularly those experiencing rapid population concentration and urban expansion. This typical structure often manifests as a multi-ring road network, with approximately five major rings surrounding the city centre. By standardising our analysis to $l=5$, we aim to create a consistent

framework for comparing urban mobility across different cities, despite the diverse names and specific configurations of their ring roads. This approach ensures that our analysis remains focused on the principle of equal trip generation, facilitating a more accurate and comparable understanding of urban mobility patterns (p.6).

We hope these revisions and clarifications address your concerns and enhance the comprehensibility of our methodology and its theoretical underpinnings.

e. Part E, the main analysis is based on Fig. 5. Yet, its presentation should be improved. For instance, putting the Insert within (c) (with a different scale) is confusing. With the three diagrams, there is perhaps no need for the Insert again. The colours are badly chosen, with late evening and near midnight (2300) as red. Please try having green and blue at the two ends of early morning and late night (suggesting nighttime) and red and yellow for the midday. Also, “Error bars represent the standard errors” is irrelevant. Can the variations be summarized statistically for comparison?

Response: We are thankful for your constructive critique concerning the presentation of Figure 5 and its interpretation, along with the colour scheme and statistical representation of the data. We acknowledge the importance of clear and intuitive visualisation for conveying our findings effectively.

Following your suggestion, we have revisited the layout of Figure 5 to enhance clarity and avoid confusion (p.9). The insert within panel (c) has been removed to streamline the presentation. Addressing the relevance of error bars and the need for a statistical summary of variations, we have incorporated additional quantitative comparisons in the supplementary materials (SI p.16). Supplementary Table 5 now presents measures of range and standard deviation to quantitatively compare the variations in anisotropy and centripetality among the three city types throughout the day. Supplementary Table 6 shows measures of change and change rate to quantitatively compare the variations in anisotropy and centripetality among the three city types during morning peak hours. This approach allows us to make substantiated statements about the temporal patterns of urban mobility in strong monocentric, weak monocentric, and polycentric cities, providing a clearer, more rigorous basis for our analysis.

3) Part III on Mechanism

a. The RWRC model is fine. Is this part purely theoretical? Do you have any idea about the empirical values of α and β across the 60 cities?

Response: We appreciate your interest in the RWRC model and your inquiry regarding its theoretical nature and the empirical values of parameters α and β across the 60 cities studied. Your question touches on a critical aspect of our research—the bridge between theoretical modeling and empirical validation.

The RWRC model, as presented in our study, is indeed theoretical and designed to establish a foundational understanding of urban mobility dynamics by focusing on the essential factors influencing workplace and residence choices. At this stage, the model primarily serves as a conceptual framework to explore the potential mechanisms underpinning observed urban mobility patterns (p.9).

Regarding the empirical values of α and β , we acknowledge the importance of grounding theoretical models in real-world data to enhance their applicability and relevance. Currently, our analysis does not extend to the direct empirical estimation of these parameters across the 60 cities due to the model's initial theoretical orientation and the complex interplay of factors influencing urban mobility that our current dataset cannot fully capture.

However, inspired by your comment, we recognise the significant potential for future research to refine the RWRC model into a more comprehensive and empirically grounded tool. By incorporating additional variables, such as traffic congestion, housing prices, and socioeconomic status, we aim to develop a model that not only aligns more closely with empirical observations but also offers actionable insights for urban planners and policymakers.

In the discussion section of our manuscript, we elaborate on this future direction, emphasising the need for an expanded model that integrates a broader range of determinants affecting mobility choices. This approach will enable a more nuanced analysis of urban mobility, bridging the gap between theoretical assumptions and the complex realities of urban living (p.11).

4) Part IV on Discussion

a. Generally fine. See overall comments above.

Response: Thanks for your comments. We have revised the paper based on your overall comments (p. 11).

5) Part V on Methods

a. The authors suggested that the dataset consists of mobile phone signal records collected over a 2-month period, representing approximately 90 million residents across 60 cities. Yet, in Part B, it seems that only data of the peak hour (7:00-7:59) of 25 weekdays were used for the cluster analysis. What proportion of the dataset does this represent? Does this only apply to Part B of Section II on Results? Make this clear in the main text.

Response: We are grateful for your astute observation and the opportunity to clarify the scope of data usage in our analysis. Commuting behaviours, particularly during peak hours, are pivotal for delineating the typical urban travel structure, hence our focus on the morning peak hour (7:00-7:59) across 25 weekdays within the two-month dataset for cluster analysis.

For each city, the morning peak hour's travel volume accounts for approximately 8% of the total daily trips (SI p.2). This specific selection aims to capture the essence of urban mobility patterns, reflecting the structured nature of urban life and its influence on mobility trends. Such an approach allows us to distil meaningful insights into the spatial directionality of urban mobility and to conduct a nuanced cluster analysis of the cities based on their commuting patterns.

This detail has been added to the supplementary materials under the section "Characterising the typical morning peak hour," where we describe the dataset and the rationale behind focusing on the peak commuting hour for our analysis (SI pp.17-18). Furthermore, we wish to clarify that this approach applies not only to Part B but also to Parts A, C, and D of Section II on Results, ensuring a consistent analytical framework across our study.

In response to your feedback, we have made explicit mentions in the main text to ensure readers are fully aware of the dataset's specific usage and the reasoning behind our methodological choices

(pp.4-6). This clarification aims to enhance the transparency and understanding of our research process, enabling a clearer interpretation of our findings.

b. Then, in Part C, it says that actually only the urban areas are considered as “origins” of human mobility flows. Does this apply to all Results reported in Section II? From Fig. 2, it seems that the “urban areas” may vary from about half to one-tenth of the “city boundary”. Again, what proportion of the dataset does this represent? Please provide in Supplementary Table 2 the size of the urban area (now listed?), the size of the city (based on city boundary), the share of the former in the latter, and the population of the urban area and city, respectively.

Response: We are grateful for your detailed inquiry regarding the focus on urban areas as the origins of human mobility flows within our analysis and the implications of this approach for the results reported in Section II. Acknowledging the importance of clarity and precision in our methodological definitions, we have indeed specified that our analysis concentrates on mobility flows originating from defined urban areas, applicable across all results presented in this section (pp.4,12).

To address your request for a more detailed breakdown of the dataset in relation to urban and city areas, we have supplemented our manuscript with additional information in Supplementary Note 3 (SI p.18). As you rightly pointed out, there is a notable variance in the proportion of urban areas relative to the total city boundary across the 60 cities in our study. This variance is crucial for understanding the spatial context of our analysis and underscores the significance of distinguishing between administrative city boundaries and the actual urban extents where human mobility patterns are most pronounced.

In Supplementary Table 1, we provide a comprehensive comparison of the sizes of urban areas and city areas as defined by administrative boundaries, including the ratio of urban to city area sizes for each city (SI p.12). Similarly, Supplementary Table 2 details the population sizes of urban areas and their respective cities, along with the share of the urban population in relation to the total city population (SI p.13). These tables illuminate the considerable variability in the relationship between administrative and actual urban extents, highlighting disparities that are critical to our analysis. The urban to city area ratio varies significantly, underscoring the concentrated nature of urban zones as hubs of human activity despite their relatively compact spatial footprint. On average, urban populations represent 40% of total city populations, further emphasising the centrality of these areas in studies of urban mobility.

By including this detailed information, we aim to provide a clearer and more comprehensive view of the dataset’s scope and the methodological considerations that underpin our analysis. We hope this additional context addresses your questions and enhances the transparency and interpretability of our findings.

6) Supplementary Figures

a. Figure 1, are the arrows between B and C in diagrams (b) and (c) correct?

Response: Thank you for bringing this to our attention. Upon reviewing Figure 1, we realised that there was indeed an error in the direction of the arrows between B and C in diagrams (b) and (c) (SI p.1). This mistake was an oversight on our part, and we have corrected the diagrams to accurately

reflect the intended direction of flows.

b. Figure 5, should (c) be (b) in the title description?

Response: Thank you very much for your attentive observation regarding the labelling in Figure 5. We have now rectified this error to ensure that the labels accurately correspond with the figure parts they describe (SI p.6).

c. Figures 11-13, is there a better way to differentiate the three groups, other than simply describing each of them as “there is a noticeable increase in anisotropy”?

Response: Thank you for your insightful suggestion regarding the presentation of differences among the three groups of cities in Figures 11-13. We acknowledge the importance of clearly communicating the nuanced variations in anisotropy among the city types and have revised our approach to differentiate these groups more distinctly.

In response to your feedback, we have introduced a new visualization strategy in Supplementary Figure 9, which now includes detailed linear regression analyses for each city type (SI p.9). This approach allows us to quantitatively distinguish between strongly monocentric, weakly monocentric, and polycentric cities based on the rate of change in anisotropy and centripetality across urban tiers. The dashed red lines in these figures represent the linear regression results, highlighting the distinct trends in anisotropy and centripetality for each group. The slope of these lines effectively differentiates the groups, illustrating the varying rates of increase in anisotropy from strong monocentric to polycentric cities (SI p.21).

This revised presentation, detailed in the supplementary materials, enriches our analysis by offering an additional perspective on the differences among city types, complementing the visual differentiation presented in the main text. By employing this analytical method, we aim to provide a clearer and more comprehensive understanding of the spatial dynamics characterising each city type.

d. Figures 14-16, is there a better way to differentiate the three groups, other than simply describing each of them as “there is a noticeable decrease in centripetality”?

Response: Thank you for your constructive suggestion regarding the presentation of changes in centripetality among the three groups of cities in Figures 14-16. Your feedback underscores the importance of providing a nuanced and clear distinction in the trends observed across different city types. In response, we have revised our approach to offer a more detailed and analytical perspective on these variations.

We have enriched the analysis presented in Supplementary Figure 9 to clearly delineate the differences in centripetality trends among strongly monocentric, weakly monocentric, and polycentric cities (SI p.9). This enhancement includes detailed linear regression results for each city type, allowing for a quantitative differentiation based on the slope of the regression lines. These lines illustrate the distinct patterns of change in centripetality, with strong monocentric cities showing the slowest decrease, followed by weak monocentric cities, and finally, polycentric cities exhibiting the most rapid decrease (SI p.21).

Additionally, we highlight an interesting observation that, in some strongly monocentric cities, centripetality increases with spatial level, reflecting the strong gravitational pull of the urban core. This trend suggests a significant attraction towards the city centre in these cities, underscoring the unique spatial dynamics at play (SI p.22).

By incorporating these detailed analyses, we aim to provide a clearer, more comprehensive understanding of how centripetality varies across city types, enhancing the interpretability of our findings. The revised presentation in the supplementary materials not only differentiates the three groups more effectively but also contributes to a deeper insight into the underlying urban mobility patterns.

e. Figures 17-19, the diagrams are a bit difficult to read. The x- and y-axes are difficult to see. Also, what do the colours represent? What do the lines represent?

Response: We deeply appreciate your valuable feedback regarding the readability of Figures 17-19. We recognise the importance of clear and comprehensible visual representations in conveying our research findings effectively. Following your suggestions, we have undertaken several measures to enhance the clarity and interpretability of these diagrams (SI p.10).

First, we have adjusted the visualisation settings to ensure that the x- and y-axes are more visible and easily discernible. This adjustment aims to facilitate a better understanding of the axes labels and the scales used, enhancing the overall readability of the diagrams.

Second, to address your query about the representation of colours and lines in the diagrams, we have added detailed legends and explanations within the figure captions. Specifically, the colours of the points now clearly correspond to different hours of the day, providing insights into temporal variations in urban mobility patterns. Additionally, we have clarified that the lines of different colours and types represent distinct urban mobility patterns observed across various city types. This distinction is crucial for interpreting the data and understanding the mobility dynamics presented in each diagram.

To further improve clarity, we have focused on selecting representative cities for each city type to illustrate in the figures. This approach aims to highlight the distinctive features and trends of strongly monocentric, weakly monocentric, and polycentric cities more effectively. By concentrating on these representative examples, we hope to provide a clearer, more focused visual narrative of urban mobility patterns.

We sincerely apologise for any initial difficulties in interpreting these figures and hope that these revisions enhance your understanding of our analysis. Your feedback has been instrumental in improving the quality of our visual presentation. Thank you for contributing to the enhancement of our manuscript.

7) Supplementary Notes

a. Supplementary Note 6, this has to be re-written. There are many unclear terms, such as “inhomogeneity” and “equal mass”. The main logic of the “spatial hierarchical structure” is also not clear. It needs to be elaborated clearly, instead of just stating that “we model urban space as a hierarchy of l levels according to the principle of equal trip occurrence”.

Response: We are immensely grateful for your feedback regarding the clarity and presentation of

Supplementary Note 6. Your detailed observations have prompted us to undertake a thorough revision of this section to ensure that it effectively communicates the underlying concepts and methodologies of our analysis.

In response to your concerns, we have taken the following steps to enhance the clarity and comprehensibility of Supplementary Note 7 (SI p.21):

We have revisited and revised the use of technical terms such as “inhomogeneity” and “equal mass” to ensure they are clearly defined and contextualised within the framework of our study. We understand that the accessibility of our supplementary notes is crucial for readers to fully grasp the methodologies employed in our analysis.

Recognising the critical importance of a clear exposition of the “spatial hierarchical structure,” we have expanded our explanation to detail the rationale and process behind modelling urban space as a hierarchy of l levels. This includes a more comprehensive discussion on the principle of equal trip occurrence that underpins this hierarchical structuring, aiming to elucidate how it facilitates a nuanced analysis of urban mobility patterns.

We have restructured Supplementary Note 7 to present a more coherent and logical progression of ideas, ensuring that the concept of a spatial hierarchical structure and its significance in our study are clearly articulated. This involves a step-by-step breakdown of how urban spaces are segmented into levels based on mobility data, emphasising the theoretical and practical implications of this approach.

Through these revisions, we aspire to provide a more lucid and detailed account of the methodologies that form the backbone of our analysis, thereby improving the overall quality and impact of our supplementary materials.

Reviewer #1 (Remarks to the Author):

The effort I see in the revised paper has addressed all my concerns. I am more than happy to see this work published as it is. Green light from me!

Reviewer #1 (Remarks on code availability):

All is good.

Reviewer #2 (Remarks to the Author):

I'm satisfied with the authors' response to my comments and the revisions they have undertaken as a result.

Reviewer #3 (Remarks to the Author):

Thanks a lot. My comments have been addressed satisfactorily.

Response to reviewer comments

Reviewer #1 (Remarks to the Author):

The effort I see in the revised paper has addressed all my concerns. I am more than happy to see this work published as it is. Green light from me!

Reviewer #1 (Remarks on code availability):

All is good.

Response: We thank the reviewer for a positive appraisal of our revised manuscript, and for recommending its publication in Nature Communications.

Reviewer #2 (Remarks to the Author):

I'm satisfied with the authors' response to my comments and the revisions they have undertaken as a result.

Response: Many thanks to the reviewer for the great reviews. The manuscript has benefited much from the suggestions.

Reviewer #3 (Remarks to the Author):

Thanks a lot. My comments have been addressed satisfactorily.

Response: We extend our gratitude to the reviewer for dedicating time to evaluate our work. The constructive comments are very helpful to strengthen this manuscript.